

# Hurricane impacts in the United States East Coast offshore wind energy lease areas

Kelsey B. Thompson[1], Rebecca J. Barthelmie[2], Sara C. Pryor[1]

[1]Department of Earth and Atmospheric Sciences, Cornell University, Ithaca, NY 14853, USA
[2]Sibley School of Mechanical and Aerospace Engineering, Cornell University, Ithaca, NY 14853, USA

*Correspondence to*: Kelsey B. Thompson (kbt32@cornell.edu)

**Abstract.** Wind turbines deployed in offshore wind energy lease areas along the U.S. East Coast could significantly contribute to the national electricity supply. This region is also impacted by powerful tropical and extra-tropical cyclones that may lead to high structural loading on wind turbines and support structures and, in the event of above cut-out wind

speeds, low power production (capacity factors < 0.2). Four sets of high-resolution simulations are performed for two category 3 tropical cyclones that tracked close to current offshore wind energy lease areas to assess the possible impacts on, and from, wind turbines. Simulations of Hurricanes Irene and Sandy are performed at convective permitting resolution with both the Weather Research and Forecasting Model (WRF) and the Coupled Ocean-Atmosphere-Wave-Sediment Transport (COAWST) Model to characterize geophysical conditions of relevance to offshore wind turbines. These simulations are

performed without and with a wind farm parameterization (WFP) active with the latter using the assumption that existing lease areas are fully populated with 15 MW wind turbines at a 1.85 km spacing. Many aspects (e.g., track, near-surface wind speed, sea level pressure, precipitation volumes) are well reproduced in control simulations (no WFP) with both WRF and COAWST particularly for Hurricane Sandy. COAWST simulations lead to more intense cyclones with a slightly larger area of storm-force wind speeds, a higher likelihood of hub-height wind speeds > 25 m s$^{-1}$, plus higher precipitation volumes,

possibly indicating under-estimation of hurricane risk in uncoupled simulations. All eight simulations indicate maximum hub-height wind speeds (HH WS) within the existing lease areas below 50 m s$^{-1}$. However, COAWST simulations indicate frequent wind-wave misalignment of > 30° and the joint occurrence of significant wave height, hub-height wind speed, and wave period in some lease areas reach levels that are likely to be associated with large structural loads. This work re-emphasizes the utility of coupled simulations in describing geophysical conditions of relevance to offshore wind turbine

operating conditions.

## 1 Introduction

### 1.1 Motivation

At the end of 2023, the global offshore wind energy installed capacity (IC) was approximately 75.2 GW (GWEC, 2024a) due in part to a 24% increase in installed capacity during 2023 (GWEC, 2024b). The plentiful offshore wind resource (Marvel et

al., 2013; Bodini et al., 2024; Pryor and Barthelmie, 2024b) and recent reductions in the Levelized Cost of Energy for offshore deployments (Jansen et al., 2020; Wiser et al., 2021) mean that the number of wind turbines (WT) deployed in (coastal) offshore regions is projected to rapidly increase (GWEC, 2024b; Pryor and Barthelmie, 2024b).

The offshore environment presents significant challenges for making long-term, climatologically representative robust

measurements of properties such as wind speed at wind turbine hub-height (HH WS) (Foody et al., 2024) that are critical for determining the wind resource and key aspects of operating conditions (IEC, 2019b, a; Mudd and Vickery, 2024). The relative paucity of measurements leads to financial uncertainty and thus potentially jeopardizes realizing U.S. national goals for achieving the energy transition (Hansen et al., 2024). It also means that numerical modeling is playing a critical role in





projecting wind resource and operating conditions in offshore wind energy development areas (Kresning et al., 2020; Pryor
and Barthelmie, 2021; Bodini et al., 2024; Pryor and Barthelmie, 2024b; Wang et al., 2024).

Substantial offshore wind energy developments are planned or in progress along the U.S. East Coast (Fig. 1) in regions with
high wind resource (power generation potential), close proximity to major demand centers, and shallow water depths (Pryor
et al., 2021; Pryor and Barthelmie, 2024b, a). This region also has the potential to be impacted by tropical cyclones and/or
transitioning tropical-extratropical cyclones (Xie et al., 2005; Baldini et al., 2016; Barthelmie et al., 2021; Wang et al.,
2024).

Wind speeds within tropical cyclones frequently exceed the threshold at which wind turbines cease power production (25-30
m s$^{-1}$) to avoid high operational loads (Petrović and Bottasso, 2014). There are reports of individual wind turbine failures
during hurricanes (Chen and Xu, 2016), and six wind turbines in a wind farm without hurricane-resistant wind turbines were
damaged by 65 m s$^{-1}$ wind speeds during Typhoon Yagi in September 2024 (Yihe, 2024). Accordingly, hurricane-induced
extreme wind conditions represent an important component of wind turbine design standards (IEC, 2019b, a; Ju et al., 2021;
Martín del Campo et al., 2021). For offshore wind farm lease areas (LA) in coastal waters along the U.S. East Coast (Fig. 1),
past research has suggested areas north of Maryland have the lowest risk of hurricane damage (< 5% probability that in a 20
year period more than 10% of the wind turbines would be destroyed) (Rose et al., 2012b, a). Offshore LA near North
Carolina experienced fewer than 40 instances of hurricane force winds (10-m wind speeds > 33 m s$^{-1}$) between 1900 and
2013, while those located near Maryland and farther northward experienced ~ 20 instances (Hallowell et al., 2018). Based on
output from the 30 km resolution ERA5 gridded dataset, the highest 50-year return period (RP) wind speed (WS) at 100 m
above sea level (a.s.l.) and significant wave heights (Hs, i.e., the mean height – crest to trough of the largest one-third of
waves) for the U.S. East Coast offshore wind energy LA are ~ 39.7 m s$^{-1}$ and ~ 11 m, respectively (Barthelmie et al., 2021).
Equivalent estimates from buoy measurements are; 32.6 m s$^{-1}$ and 9.5 m (Kresning et al., 2024). A further model-based study
indicated 50-year RP Hs of 9-11 m for the northernmost LA considered here (McElman et al., 2024).

Wind-wave coupling plays a key role in both near-surface atmospheric processes and wind turbine loading (Valamanesh et
al., 2013; Valamanesh et al., 2015; Koukoura et al., 2016; Hallowell et al., 2018; Hashemi et al., 2021; Li et al., 2022; Müller
et al., 2024). Analyses based on buoy-based measurements of wind and waves along the U.S. East Coast indicated the mean
failure probability during a 20-year wind turbine lifetime is $9.6 \times 10^{-6}$ for a functional wind turbine yaw control system and
$2.9 \times 10^{-4}$ for a non-functional yaw control system (Hallowell et al., 2018). Wind-wave directional offset is also considered
in offshore wind turbine design codes (IEC61400-3) for loading on the support structure (IEC, 2019a). A 90° wind-wave
misalignment is projected to increase the mud-line bending moment for a monopile foundation by up to a factor of five, and
even more modest misalignment of 30° approximately doubles this bending moment (Fischer et al., 2011). This
amplification of bending moment with wind-wave misalignment is greatly enhanced under high HH WS (Stewart and
Lackner, 2014).






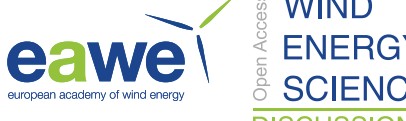

**Figure 1: Hurricane Irene (left) and Hurricane Sandy (right). (a, b) Hurricane tracks from model simulations presented herein (WRF without the action of wind turbines, WRF with the action of wind turbines, COAWST without the action of wind turbines, and COAWST with the action of wind turbines), derived from minimum SLP every 10 min and the corresponding NHC "best track" locations. HU indicates hurricane stage, TS indicates tropical storm stage, and EX indicates extratropical stage. Numbers in brackets represent the location at the times specified in Table 1. The time series plots are the 1-h precipitation volume within**
**375 km from the track location for each simulation and IMERG V07. (c, d) Accumulated IMERG V07 precipitation during the entire simulation period (1200 UTC on 24 August 2011 through 1200 UTC on 29 August 2011 and 1200 UTC on 25 October 2012 through 1200 UTC on 1 November 2012, respectively) and NHC best track locations (from (a) and (b)). Also shown are the U.S. state abbreviations; Maine (ME), New Hampshire (NH), Vermont (VT), Massachusetts (MA), Rhode Island (RI), Connecticut (CT), New York (NY), New Jersey (NJ), Pennsylvania (PA), Ohio (OH), West Virginia (WV), Maryland (MD), Delaware (DE),**
**Virginia (VA), North Carolina (NC), and South Carolina (SC). (e, f) Difference in accumulated precipitation (WRF without the action of wind turbines minus IMERG) and the hurricane track from the WRF no WT simulation (every 10 min). (g, h) Difference in accumulated precipitation (COAWST without the action of wind turbines minus IMERG) and the hurricane track from the COAWST no WT simulation (every 10 min).**

**Table 1: The time (UTC) and date that correspond to the numbered locations for Hurricanes Irene and Sandy shown in Fig. 1. All dates for Irene are in 2011 and all dates for Sandy are in 2012.**

| location | Irene time and date | Sandy time and date |
|---|---|---|
| **1** | 1800 27 Aug | 0000 29 Oct |
| **2** | 0000 28 Aug | 0600 29 Oct |
| **3** | 0600 28 Aug | 1200 29 Oct |
| **3a** | 0935 28 Aug (landfall at Brigantine Island, NJ) | - |
| **4** | 1200 28 Aug | 1800 29 Oct |
| **4a** | 1300 28 Aug (landfall at Coney Island, NY) | 2100 29 Oct (downgraded to extratropical) |
| **4b** | - | 2330 29 Oct (landfall at Brigantine Island, NJ) |
| **5** | 1800 28 Aug | 0000 30 Oct |
| **6** | 0000 29 Aug | 0600 30 Oct |
| **7** | - | 1200 30 Oct |
| **8** | - | 1800 30 Oct |
| **9** | - | 0000 31 Oct |
| **10** | - | 0600 31 Oct |
| **11** | - | 1200 31 Oct |
| **12** | - | 1800 31 Oct |

Estimation of design criteria extreme wind speeds from numerical modeling is critically dependent on the grid spacing at
which the model is applied (Larsén et al., 2012) and momentum dissipation at the ocean surface which in turn is determined by wind-wave coupling and the parameterization used to dictate the surface roughness length (Larsén et al., 2019; Wang et al., 2024). The Coupled Ocean-Atmosphere-Wave-Sediment Transport (COAWST) modeling system (Warner et al., 2010) comprises a series of linked model components. In the research presented herein these models are; the Weather Research and Forecasting Model (WRF) (Skamarock et al., 2019), the Regional Ocean Modeling System (ROMS) (Shchepetkin and
McWilliams, 2005; Haidvogel et al., 2008; Shchepetkin and McWilliams, 2009), and Simulating Waves Nearshore (SWAN) (Booij et al., 1999) and they interact through use of the Model Coupling Toolkit (MCT) (Jacob et al., 2005; Larson et al., 2005; Warner et al., 2008).





Only limited previous research has sought to quantify the degree to which wind-wave coupling improves simulation fidelity
and/or intensity for wind speeds at heights of relevance to offshore wind turbines. One such study focused on 23 intense
cyclones in the North Sea and found that when WRF is coupled to SWAN through a wave boundary layer model with an
innermost domain with grid spacing of 2 km, the inferred 50-year RP wind speeds were systematically higher than those
from WRF alone and the degree of agreement in extreme wind speeds at five offshore and/or coastal masts was improved
(Larsén et al., 2019). A further study found that for Tropical Storm Ana in the mid-Atlantic Bight, two-way coupled WRF
and WaveWatch III (WW3) simulated peak 90-m wind speeds were a closer match to observations than the corresponding
values from either a standalone WRF or one-way coupled WRF simulation (Gaudet et al., 2022). Simulation of Hurricane
Sandy with WRF with FVCOM (the unstructured-grid, Finite-Volume Community Ocean Model) coupled through the Earth
System Model Framework also found improved agreement with observations for the central pressure location and intensity
plus 10-m wind speeds relative to simulation solely with WRF (Li and Chen, 2022). COAWST (configured with WRF 3.2,
ROMS 3.3, and SWAN 40.81) coupled with MCT showed "modest improvement in track but significant improvement in
intensity …. versus uncoupled (e.g., standalone atmosphere, ocean, or wave) model simulations" for Hurricane Ivan
(Zambon et al., 2014a). Thus, there is provisional evidence that, in accord with expectations, detailed coupling of the
atmosphere-wave and ocean models improves simulation of atmospheric parameters within these extreme events relative to
simulations with WRF alone.


In addition to the importance of intense cyclones (e.g., hurricanes) to wind turbine design standards, there have also been
suggestions that very wide-spread deployment of offshore wind turbines in the U.S. coastal zone could aid in reducing the
intensity of tropical cyclones and thus reduce damage onshore (Jacobson et al., 2014). Simulations of three hurricanes using
the GATOR-GCMOM model indicated that offshore wind turbine arrays comprising 110,000 and 420,000 wind turbines
(installed capacities > 300 GW) at installed capacity densities (ICD) of $8 - 17$ MW km$^{-2}$ might reduce 15-m wind speeds by
over 25 m s$^{-1}$ and reduce storm surge by up to 79% (Jacobson et al., 2014). Simulations with 22,000 to 74,000 wind turbines
also suggested that offshore wind turbines could reduce the amount of precipitation over land, downstream of the wind farms
(Pan et al., 2018).

### 1.2 Objectives

Research presented herein uses storyline simulations of two of the most powerful hurricanes that have occurred within the
U.S. East Coast coastal waters in which offshore wind energy LA have been auctioned (Fig. 1, see further details in Figs. S1-
S2). Four sets of simulations are performed for each of these hurricanes; (a) WRF, (b) WRF with the action of wind turbines
included in offshore wind energy lease areas purchased prior to mid-2023, (c) COAWST, and (d) COAWST with the action
of wind turbines included. Our specific research questions are as follows:


1) Are the characteristics of these hurricanes well captured using either the WRF or COAWST models? A sub-
component of this question is does the more explicit coupling in COAWST improve simulation fidelity?

2) Do these simulations suggest that either of these hurricanes would have been characterized by either (a) widespread
loss of power production across these lease areas due to cut-out at high wind speeds and for how long and/or (b)
exceedance of wind turbine design wind speeds and/or very high wind-wave structural loading? Again, a sub-
component of this question is does use of COAWST versus WRF change wind speed intensity and/or the duration
of time with low power production?



3)  If wind turbine rotor extraction of momentum is simulated using a wind farm parameterization in WRF and COAWST, is there evidence of weakening of the hurricanes for wind turbine numbers and installed capacity densities that are likely to be achieved using the offshore wind energy lease areas considered here?

**2 Data and Methods**

**2.1 Characteristics of the hurricanes considered herein**

Research presented herein focuses on two recent hurricanes:

1)  Hurricane Irene became a category 3 hurricane, with 54 m s$^{-1}$ wind speeds at 10-m height in the Bahamas at 1200 UTC on 24 August 2011 (Avila and Cangialosi, 2011). It made landfall at Cape Lookout, North Carolina at 1200 UTC on 27 August with 39 m s$^{-1}$ 10-m wind speeds. After moving out over the water, it again made landfall, this

160        time as a tropical storm, with 31 m s$^{-1}$ wind speeds reported at Brigantine, New Jersey at 0935 UTC 28 August. The cyclone then moved over Coney Island, New York with 28 m s$^{-1}$ wind speeds reported at 1300 UTC. Simulations presented herein are initialized at 1200 UTC on 24 August 2011 and run through 1200 UTC on 29 August 2011.

2)  Hurricane Sandy became a category 3 hurricane, with 51 m s$^{-1}$ wind speeds at 10-m height in eastern Cuba at 0525

165        UTC on 25 October 2012 (Blake et al., 2013; Lackmann, 2015). It grew to have a roughly 1611 km diameter of tropical-storm-force wind speeds, before making landfall near Brigantine, New Jersey as a post-tropical cyclone with 36 m s$^{-1}$ 10-m wind speeds and a minimum pressure of 945 hPa at 2330 UTC 29 October. Simulations presented herein run from 1200 UTC on 25 October 2012 through 1200 UTC on 1 November 2012.

**2.2 Modeling**

The source of initial and lateral boundary conditions (Khaira and Astitha, 2023) and specific model configurations employed within WRF and COAWST (including the coupling system) have a critical impact on simulated flow conditions (Mooney et al., 2019). In this research, both WRF (v4.2.2) and COAWST (v3.7 and MCT v2.6.0) simulations use two domains (Fig. 2a) and the coupling interval in COAWST is 10 min (Fig. 2b). The source of boundary and initial conditions and key physics options (Tables 3 and 4) are informed by previous simulations of Hurricanes Sandy (Zambon et al., 2014b) and Irene

(Mooney et al., 2016). The MYNN2 planetary boundary layer scheme is used due to the compatibility with the Fitch wind-farm parameterization (WFP) (Fitch et al., 2012) that is used here to compute power production, momentum extraction, and turbulent kinetic energy (TKE) induced by the action of wind turbines. Following previous research (Pryor and Barthelmie, 2024b, a), we assume that all auctioned offshore lease areas along the U.S. East Coast shown in Fig. 2a are populated with IEA reference 15 MW wind turbines (Fig. 2c) with a spacing of 1.85 km for an average ICD of 4.3 MW km$^{-2}$. This results in

a total of 2642 wind turbines (Fig. 2a), each of which has a hub height of 150 m, and a rotor diameter of 240 m. The 1.85 km wind turbine spacing and 1.33 km WRF domain 02 (d02) grid spacing (dx) results in 2641 grid cells with at least one wind turbine; one grid cell has two wind turbines. Of the 71 unstaggered WRF vertical levels, level 15 has a mean height of 155 m in grid cells with wind turbines and is therefore used for HH WS. Note the wind speeds output from d02 are for a nominal model time step of 2 s but are representative of a spatial average of 1.33 km by 1.33 km, while the design standards are for a

sustained wind speed at a point (Larsén and Ott, 2022).

Variation of wave state and surface roughness length with wind speed is an important determinant of extreme, near surface wind speeds and turbulence intensity (Zambon et al., 2014a; Porchetta et al., 2019; Porchetta et al., 2020; Porchetta et al., 2021; Wang et al., 2024). The COAWST simulations are configured using the Taylor Yelland formulation (Taylor and



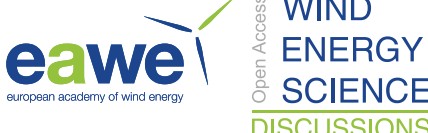

Yelland, 2001) to calculate surface roughness length following past research (Zambon et al., 2014b) that found use of this
parameterization resulted in better fidelity for Hurricane Sandy track, intensity, sea surface temperatures, and wave heights
than alternatives (Oost et al., 2002; Drennan et al., 2005). Use of the MYNN surface layer with WRF and the
DRAGLIM_DAVIS drag limiter option with COAWST, means all simulations implement a maximum ocean roughness drag
coefficient of $2.85 \times 10^{-3}$, consistent with research that has shown asymptotic behavior of drag at high wind speeds (Davis et

al., 2008). Data are output every 10 min and each simulation is subject to a warm restart every 6 h due to wall clock
limitations on the compute platform.

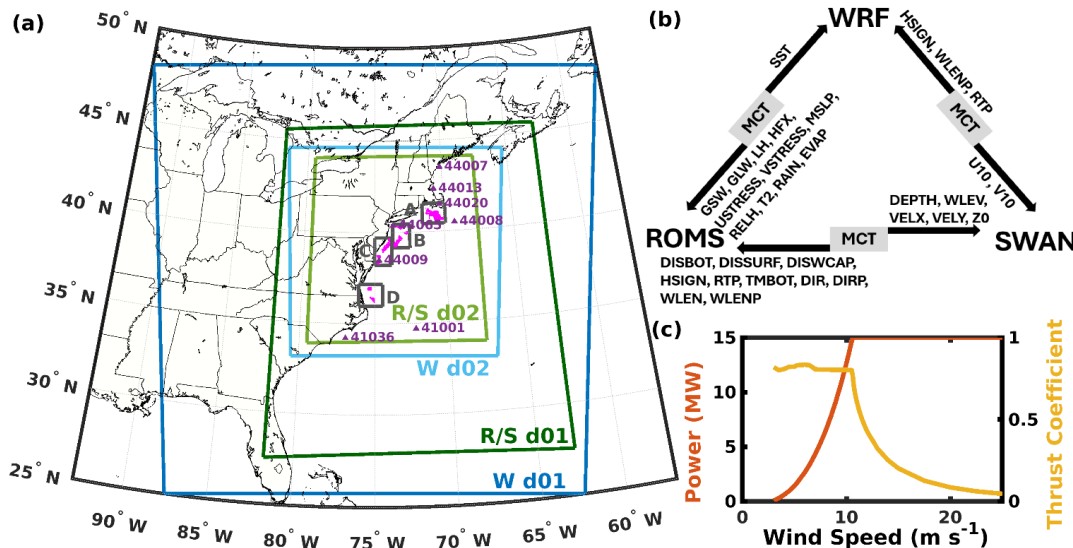

**Figure 2: (a) Domains 01 and 02 (d01 and d02) for WRF (W), ROMS (R), and SWAN (S) and locations of offshore wind energy LA**
**considered herein (in the simulations with wind turbines, these LA contain 2642 wind turbines which are shown in magenta). Also**
**shown are the eight NDBC buoys used in the simulation evaluation. (b) Schematic of information flow among the COAWST model**
**components. See Table 2 for additional details. (c) Power and thrust coefficient for the 15 MW IEA reference wind turbine**
**(Gaertner et al., 2020).**






**Table 2: Additional details about the information flow among the COAWST model components in Fig 2b.**

| name | abbreviation | units |
|---|---|---|
| depth below mean sea level | DEPTH | m |
| mean wave direction | DIR | rad |
| peak wave direction | DIRP | rad |
| energy dissipation due to bottom friction | DISBOT | $W\ m^{-2}$ |
| energy dissipation due to surf breaking | DISSURF | $W\ m^{-2}$ |
| energy dissipation due to white-capping | DISWCAP | $W\ m^{-2}$ |
| evaporation rate | EVAP | $kg\ m^{-2}\ s^{-1}$ |
| downward long wave flux at ground surface | GLW | $W\ m^{-2}$ |
| net short wave flux at ground surface | GSW | $W\ m^{-2}$ |
| upward heat flux at the surface | HFX | $W\ m^{-2}$ |
| significant wave height | HSIGN | m |
| latent heat flux at the surface | LH | $W\ m^{-2}$ |
| mean sea level pressure | MSLP | mb |
| rainfall rate | RAIN | $kg\ m^{-2}\ s^{-1}$ |
| relative humidity | RELH | - |
| relative peak period | RTP | s |
| sea surface temperature | SST | K |
| temperature at 2 m | T2 | °C |
| near bottom wave period | TMBOT | s |
| x-wind component at 10 m | U10 | $m\ s^{-1}$ |
| surface u-stress | USTRESS | $N\ m^{-2}$ |
| y-wind component at 10 m | V10 | $m\ s^{-1}$ |
| current velocity component in x direction | VELX | $m\ s^{-1}$ |
| current velocity component in y direction | VELY | $m\ s^{-1}$ |
| surface v-stress | VSTRESS | $N\ m^{-2}$ |
| mean wave length | WLEN | m |
| peak wave length | WLENP | m |
| water level | WLEV | m |
| roughness length | Z0 | m |





**Table 3: Sources of initial and boundary conditions for WRF and COAWST.**

| WRF (version 4.2.2) | |
|---|---|
| Atmosphere: | North American Mesoscale Forecast System (NAM; 12 km, 6 h) |
| Sea Surface Temperature: | Group for High Resolution Sea Surface Temperature (GHRSST) Level 4 Operational Sea Surface Temperature and Sea Ice Analysis (OSTIA) (OSTIA-UKMO-L4-GLOB-v2.0; 0.05°, 24 h) |
| Horizontal resolution: | 4 km for d01, 1.33 km for d02 |
| Model top / # vertical levels: | 50 hPa / 72 |
| Time step (dt): | 6 s for d01, 2 s for d02 |
| ROMS (version 3.9) | |
| Coastlines: | Global Self-consistent, Hierarchical, High-resolution Geography Database (GSHHG; full resolution) |
| Bathymetry (also for SWAN): | General Bathymetric Chart of the Oceans (GEBCO) 2022 (15 arc-second) |
| 3D boundary conditions, initial conditions, and climatology: | HYbrid Coordinate Ocean Model (HYCOM GLBa0.08 expt 90.9) |
| 2D boundary conditions (tides): | Advanced Three-Dimensional Circulation Model (ADCIRC 2001v2e) |
| Horizontal resolution: | 10 km for d01, 3.33 km for d02 |
| Time step (baroclinic / barotropic): | 1.5 s (d01), 0.5 s (d02) / 30 (d01 & d02) |
| # vertical levels | 25 |
| SWAN (version 41.31) | |
| Wind forcing: | Global Forecast System (GFS): 0.5°, 6 h |
| Boundary conditions: | parametric forcing file (TPAR): 30 min WaveWatch III (WW3) data |
| Horizontal resolution: | 10 km for d01, 3.33 km for d02 |
| Time step: | 12 s for d01, 4 s for d02 |
| Frequency range: | 0.04 to 1.0 |


**Table 4: Physics settings used with WRF and COAWST simulations.**

| Model physics | Key reference(s) |
|---|---|
| microphysics: WRF single-moment 6-class scheme (WSM6) | (Hong and Lim, 2006) |
| longwave radiation: Rapid Radiative Transfer Model (RRTM) | (Mlawer et al., 1997) |
| shortwave radiation: Dudhia scheme (MM5) | (Dudhia, 1989) |
| surface layer: MYNN | (Olson et al., 2021) |
| land surface: Unified Noah land surface model | (Chen and Dudhia, 2001b, a; Ek et al., 2003; Tewari et al., 2004) |
| planetary boundary layer: Mellor-Yamada Nakanishi and Niino Level 2.5 (MYNN2) | (Nakanishi and Niino, 2006) |
| cumulus param.: Kain-Fritsch (d01; none for d02) | (Kain, 2004) |
| wind farm param. (both domains) | (Fitch et al., 2012) |



**2.3 Evaluation data sets**

Critical aspects of the WRF and COAWST simulations without the action of wind turbines are evaluated using:

(i)   National Hurricane Center (NHC) "best track" information and wind radii maximum extent data from Tropical Cyclone Reports (Avila and Cangialosi, 2011; Blake et al., 2013) and the Atlantic hurricane database (HURDAT2; (Landsea and Franklin, 2013)).

(ii)  30-minute precipitation at 0.1 degree resolution from the Integrated Multi-satellitE Retrievals for the Global Precipitation Measurement (GPM) mission (IMERG) V07 final run data set (Huffman et al., 2024).

(iii) National Data Buoy Center (NDBC) buoy-based measurements of wind speeds (WS), sea level pressure (SLP), sea surface temperature (SST), and significant wave height (Hs) (NDBC, 2009) (https://www.ndbc.noaa.gov) (see Fig. 2a). Note: The eight buoys from which data are presented are a mixture of 3-meter foam and 3-meter discus buoys. The anemometer and barometer heights vary between 3.8 and 4.1 m, and 2.4 and 3.4 m.

**2.4 Analysis approach**

Hurricane centroid locations are computed every 10 minutes as the minimum SLP after 3×3 smoothing is applied to the model output and are used for comparison with the NHC best track information. The initial tracking position is the first time step when the minimum SLP is fully within d02 and tracking continues until the implied translational speed/direction of motion between adjacent time steps is inconsistent with physical expectations in terms of direction or translational speed. Hurricane Irene is tracked for 33 h, from 1800 UTC 27 August 2011 through 0300 UTC 29 August 2011 and Hurricane Sandy is tracked for 67 h, from 2300 UTC 28 October 2012 through 1800 UTC 31 October 2012 (Fig. 1). Evaluation relative to SLP and wind speed data from the NDBC buoys is performed using a search area of 3×3 grid cells. Evaluation of simulated precipitation relative to IMERG over all of d02 is performed after regridding output to the IMERG grid. The volume of water exhausted as precipitation from the tropical cyclone is computed using a search radius of 375 km (see examples in Fig. 3) around the cyclone centroid in the simulation output and from the best track locations applied to IMERG.

When evaluating hurricane impacts on wind turbines within the U.S. East Coast LA, analyses are presented for both all 2641 grid cells containing wind turbines, and four LA clusters defined as in (Pryor and Barthelmie, 2024b) and listed from north to south; A (1073 WT), B (662 WT), C (624 WT), and D (283 WT) (Fig. 2a). Because prior research has indicated the challenges in perfectly reproducing hurricane tracks, we consider conditions in both grid cells containing wind turbines and/or all ocean-based grid cells within the respective LA cluster area. To facilitate comparison across the LA clusters, power production computed by the Fitch WFP is used to compute capacity factors (CF), which are the ratio of the power produced divided by that which would be produced if all wind turbines were operating at rated capacity (15 MW). System-wide CF < 0.2 are used here as an indicator of low power production.

Three-dimensional and joint occurrences of HH WS, Hs, and peak period (Tp) in WT-containing grid cells from the COAWST simulations are presented along with histograms of estimated wind-wave misalignment at the LA cluster centers in HH WS classes of 3 – < 10.6 m s$^{-1}$, 10.6 – 25 m s$^{-1}$, and > 25 m s$^{-1}$, to represent high thrust, moderate thrust, and above rated wind speeds (Fig. 2c).





Three metrics are used to analyze the impact of wind turbines on hurricane intensity and are compared for simulations with
WRF and COAWST without and with the WFP active. The cumulative volume of precipitation within 375 km of the
minimum SLP and the mean wind speed at 500 hPa (approximately the level of non-divergence, e.g., (Riehl and Malkus,
1961)) computed for grid cells that lie 50-375 km from the centroid (i.e., beyond the likely eye radius) (Morin et al., 2024;
Müller et al., 2024) are used as metrics of intensity. The mean outermost radius of tropical storm force ($R_{18}$, 18 m s$^{-1}$, see

Fig. 3) wind speeds at 10-m computed using azimuth sectors of 10° (Powell and Reinhold, 2007) for all sectors where the
distance from the cyclone centroid to the d02 boundary is ≥ 200 km is used as a measure of cyclone size. Mood's test
(Hettmansperger and Malin, 1975) is used to assess the statistical significance of differences in the median values of these
metrics.




Figure 3: Simulated precipitation from WRF (a, c) and COAWST (b, d) during two example 10 minute periods (background color) and contours of HH WS at 25, 35, and 45 m s⁻¹ for 0900 UTC 28 August 2011 (a, b) and 1900 UTC October 2012 (c, d) when the hurricanes are close to wind turbine LA. Magenta rings mark 50 km and 375 km from the minimum SLP. The black rings
mark $R_{18}$ radii of (a) 274 km, (b) 301 km, (c) 539 km, and (d) 541 km. For legibility, the colorbar is truncated. Maximum precipitation is (a) 14.0 mm, (b) 13.8 mm, (c) 18.4 mm, and (d) 16.8 mm.

## 3 Results

### 3.1 Evaluation of the no-wind turbine simulations

Simulations of Hurricane Irene exhibit lower fidelity than those of Hurricane Sandy. The centroid of Hurricane Irene is consistently displaced west (further inland, Fig. 1a) than the NHC best track data, and the translational speed is also negatively biased in simulations with both WRF and COAWST. This bias is consistent with previous COAWST simulations of this hurricane performed with 12 km grid spacing and using a range of initial and lateral boundary conditions (Mooney et al., 2019). Simulation bias is also evident in comparison with buoy observations (Table 5, Fig. 4, and Figs. S3-S9) both in
terms of the magnitudes and timing of the maximum wind speeds and minimum SLP. The maximum near-surface wind speeds differ (model minus buoy observations) by between -4.2 and 0.7 m s⁻¹ (WRF at 2.6 m) and -3.2 and 6.0 m s⁻¹ (COAWST at 2.6 m). The displacement of the Hurricane Irene centroid in the simulations results in higher over land precipitation (by up to 209 mm in some IMERG grid cells) and negative bias offshore (Fig. 1). However, the volume of water vented from the hurricane is relatively well reproduced in the simulations. During the time of tracking, the mean (1-h)
precipitation volume within 375 km of the centroid is $9.87 \times 10^8$ m³ based on IMERG combined with the NHC best track data, while the corresponding values (and percent error) are $1.14 \times 10^9$ m³ (15.1% overestimation) and $1.18 \times 10^9$ m³ (19.9% overestimation), for the WRF and COAWST simulations, respectively (Fig. 1). Mean $R_{18}$ is 279 km in the WRF simulation but is larger by an average of 23 km in 192 of the 199 tracked 10-min positions in the COAWST simulation. These mean $R_{18}$ values are similar to, but smaller than, those reported at a 6-hrly interval from HURDAT2 based on analyses in four
quadrants (mean $R_{18}$ of 495 km in the SE quadrant and 157 km in the NW quadrant). The differences in $R_{18}$ and precipitation in theses simulations versus HURDAT2 and IMERG are likely due to bias introduced by proximity to the d02 boundary in the simulation and differences in the fraction of the system over land (Chen and Yau, 2003) due to differences in storm tracks (Fig. 1).

As in past research (Zambon et al., 2014b), both the WRF and COAWST simulations of Hurricane Sandy exhibit good agreement with NHC best track data up to about 12 h after landfall in New Jersey (Fig. 1b). The mean distance between the centroids for locations; 1, 2, 3, 4, 4a, 4b, and 5 in Fig. 1b is 66.7 km (WRF) and 51.4 km (COAWST). For those seven times the distances of separation range from 30.3 to 129.1 km (WRF) and 12.0 to 71.1 km (COAWST). These positional offsets are smaller than those presented in previous research on, for example, cyclonic storm Ockhi (Mukherjee and Ramakrishnan,
2022) and the mean values from Tropical Storm Delta, Hurricane Ophelia, Hurricane Leslie, and Tropical Storm Theta (Calvo-Sancho et al., 2023). Consistent with expectations, agreement tends to degrade once the cyclone has made landfall as the system becomes less organized and more asymmetric (Zambon et al., 2014b). Modeled time series of SLP, SST, Hs, and wind speeds exhibit some level of agreement with the NDBC buoy observations in terms of time-variability (Table 6, Fig. 4, and Figs. S3-S9). As expected, due to spatial averaging and the difference in height, the maximum wind speeds from the
lowest model level (~ 2.6 m) are generally lower than the point observations on the buoys at 3.8 or 4.1 m. Seven of the eight values are smaller in the WRF simulation, and six of eight comparisons indicate lower values in the COAWST simulation. The maximum near-surface wind speeds (model minus buoy observations) differ by 3.1 to 9.0 m s⁻¹ (WRF) and 2.9 to 7.8 m s⁻¹ (COAWST). The mean absolute difference in minimum SLP is 3.1 hPa (WRF) and 2.6 hPa (COAWST). Significant wave





heights from COAWST are generally lower than those from the buoy observations, with a (model minus buoy observations)

difference of -3.7 to 0.1 m (a bias of up to 37%). The spatial pattern of precipitation from the simulations exhibits some similarity with IMERG, although the centroid of the region of maximum precipitation is displaced westward (towards the coast of North Carolina) in IMERG relative to the WRF and COAWST simulations by approximately 125 km (Fig. 1). The volume of precipitation vented from the hurricane exhibits relatively good agreement between the simulations and IMERG. The mean (1-h) precipitation volume within 375 km from the centroid is $5.35 \times 10^8$ m$^3$ based on IMERG combined with the

NHC best track data, while the corresponding values (and percent error) are $5.82 \times 10^8$ m$^3$ (+8.7%) and $5.83 \times 10^8$ m$^3$ (+8.9%) for the WRF and COAWST simulations (Fig. 1). Mean $R_{18}$ prior to landfall is 506 km in the WRF simulation and is larger by an average of 14 km in 119 of the 151 tracked 10-min positions in the COAWST simulation. The $R_{18}$ estimates from 6-h HURDAT2 data before and after landfall are 705 km and 776 km and are larger than those from the WRF and COAWST simulation due to biases in the calculation when the hurricane extends beyond the d02 boundary.


**Table 5: Comparison of WRF and COAWST output (the simulations without wind turbines) and buoy measurements. The magnitude and time (in UTC) of the maximum wind speed (WS), maximum significant wave height (Hs), and minimum SLP are given for each buoy and simulations with WRF and COAWST (simulated WS at two heights, 10 m | 2.6 m). All the times with Irene are in August 2011. With the simulations, magnitudes are provided every 10 min. With the buoys, magnitudes are provided**
**every 10 min for WS and provided at 50 min past the top of the hour for Hs and SLP.**

| Irene | | max WS (m s$^{-1}$) | time max WS | max Hs (m) | time max Hs | min SLP (hPa) | time min SLP |
|---|---|---|---|---|---|---|---|
| 41001 | buoy | 19.7 | 1840 & 1850 27 Aug | 10.0 | 1550 27 Aug | 997.5 | 1850 27 Aug |
| | WRF | 18.3 \| 16.0 | 0150 28 Aug | - | - | 1002.0 | 1020 28 Aug |
| | COAWST | 20.2 \| 17.9 | 0540 28 Aug | 5.1 | 0500 28 Aug | 1002.1 | 1040 28 Aug |
| 41036 | buoy | 25.1 | 1620 27 Aug | 8.6 | 0550 27 Aug | 957.1 | 0950 27 Aug |
| | WRF | 27.1 \| 22.9 | 2240 27 Aug | - | - | 965.3 | 2340 27 Aug |
| | COAWST | 34.4 \| 31.1 | 2340 27 Aug | 7.6 | 2040 27 Aug | 958.2 | 2210 27 Aug |
| 44007 | buoy | 16.5 | 1930 28 Aug | 4.5 | 2150 28 Aug | 983.2 | 0150 29 Aug |
| | WRF | 22.5 \| 15.6 | 0700 29 Aug | - | - | 977.5 | 0810 29 Aug |
| | COAWST | 23.3 \| 19.1 | 1140 \| 1150 29 Aug | 5.5 | 0700 29 Aug | 975.6 | 0810 29 Aug |
| 44008 | buoy | 18.3 | 1750 28 Aug | 8.2 | 1750 28 Aug | 996.1 | 1950 28 Aug |
| | WRF | 20.0 \| 14.1 | 0700 29 Aug | - | - | 997.7 | 0420 29 Aug |
| | COAWST | 22.0 \| 15.1 | 0550 \| 0250 29 Aug | 7.9 | 0650 29 Aug | 997.1 | 0300 29 Aug |
| 44009 | buoy | 21.4 | 2140 27 Aug | 6.4 | 0450 28 Aug | 958.3 | 0650 28 Aug |
| | WRF | 27.9 \| 22.1 | 1330 \| 1710 28 Aug | - | - | 974.0 | 1700 28 Aug |
| | COAWST | 28.6 \| 23.3 | 1330 \| 1400 28 Aug | 6.9 | 1410 28 Aug | 976.5 | 1610 28 Aug |
| 44013 | buoy | 18.8 | 1830 28 Aug | 3.8 | 1550 28 Aug | 984.0 | 2050 28 Aug |
| | WRF | 24.7 \| 17.1 | 0500 29 Aug | - | - | 979.2 | 0540 29 Aug |
| | COAWST | 22.7 \| 15.7 | 0410 29 Aug | 3.5 | 0420 & 0430 29 Aug | 978.3 | 0610 29 Aug |
| 44020 | buoy | 21.2 | 2350 28 Aug | 2.4 | 1550 28 Aug | 989.2 | 2050 28 Aug |
| | WRF | 26.7 \| 18.5 | 0600 \| 0840 29 Aug | - | - | 987.2 | 0600 29 Aug |
| | COAWST | 28.0 \| 18.9 | 0520 29 Aug | 3.6 | 0410 29 Aug | 985.9 | 0520 29 Aug |
| 44065 | buoy | 21.1 | 1220 28 Aug | 8.0 | 1250 28 Aug | 968.0 | 1250 28 Aug |
| | WRF | 26.6 \| 19.7 | 2030 28 Aug \| 0310 29 Aug | - | - | 974.9 | 2350 28 Aug |
| | COAWST | 27.7 \| 23.9 | 0350 29 Aug | 5.5 | 2110 28 Aug | 973.7 | 2240 28 Aug |




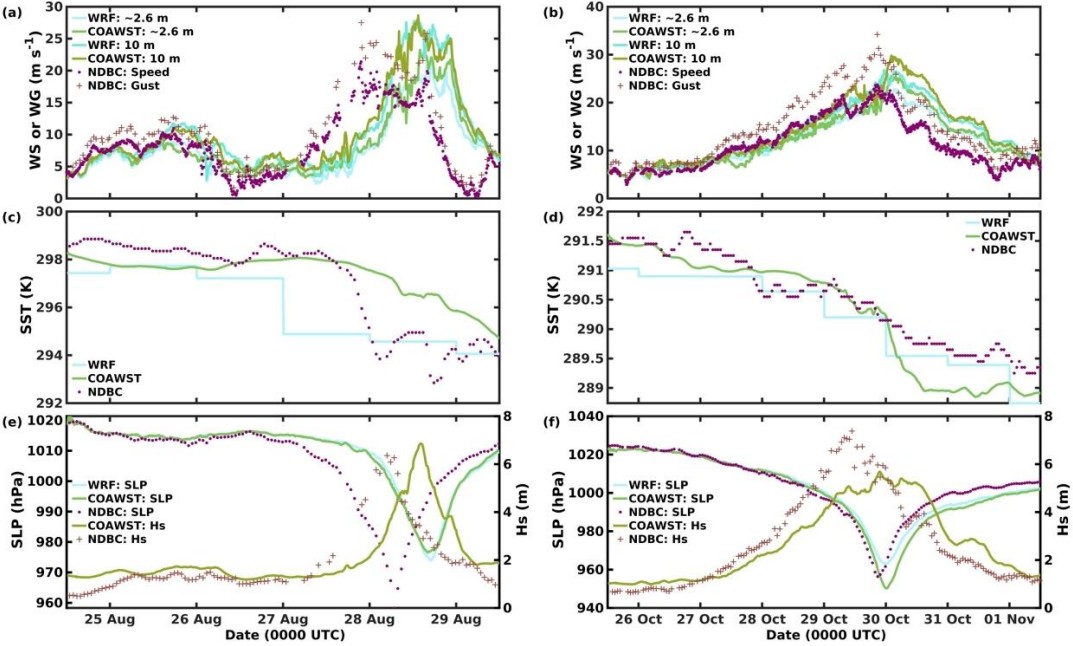


**Figure 4: Time series of (a, b) wind speeds and wind gusts (WS or WG [m s⁻¹]), (c, d) sea surface temperatures (SST [K]), and (e, f) sea level pressure (SLP [hPa]) and significant wave height (Hs [m]) from observations and simulations (WRF and COAWST) for Hurricane Irene (a, c, and e) and Hurricane Sandy (b, d, and f) at buoy 44009 (see location in Fig. 2a). Simulations performed without the action of wind turbines.**








**Table 6: Comparison of WRF and COAWST output (the simulations without wind turbines) and buoy measurements. The magnitude and time (in UTC) of the maximum wind speed (WS), maximum significant wave height (Hs), and minimum SLP are given for each buoy and simulations with WRF and COAWST (simulated WS at two heights, 10 m | 2.6 m). All the times with Sandy are in October 2012. With the simulations, magnitudes are provided every 10 min. With the buoys, magnitudes are provided every 10 min for WS and provided at 50 min past the top of the hour for Hs and SLP.**


| Sandy | | max WS (m s⁻¹) | time max WS | max Hs (m) | time max Hs | min SLP (hPa) | time min SLP |
|---|---|---|---|---|---|---|---|
| **41001** | *buoy* | 28.4 | 0610 29 Oct | 10.1 | 1350 29 Oct | 969.6 | 2350 28 Oct |
| | *WRF* | 37.4 \| 32.2 | 0630 29 Oct | - | - | 962.8 | 0250 29 Oct |
| | *COAWST* | 36.2 \| 31.8 | 0630 29 Oct | 9.8 | 1520 28 Oct | 964.3 | 0200 29 Oct |
| **41036** | *buoy* | 21.7 | 2250 27 Oct | 5.7 | 2050 & 2150 27 Oct | 992.1 | 0950 28 Oct |
| | *WRF* | 25.0 \| 20.9 | 0720 28 Oct | - | - | 995.3 | 1300 28 Oct |
| | *COAWST* | 24.6 \| 21.5 | 0800 28 Oct | 5.1 | 0630 28 Oct | 995.0 | 1530 28 Oct |
| **44007** | *buoy* | 18.0 | 0050 30 Oct | 7.1 | 0350 30 Oct | 995.9 | 0050 30 Oct |
| | *WRF* | 23.8 \| 16.6 | 0150 30 Oct | - | - | 995.6 | 0140 30 Oct |
| | *COAWST* | 23.6 \| 16.6 | 0020 30 Oct \| 2350 29 Oct | 6.6 | 0230 30 Oct | 995.6 | 0330 30 Oct |
| **44008** | *buoy* | 22.4 | 1640 29 Oct | 11.0 | 2050 29 Oct | 981.2 | 1750 29 Oct |
| | *WRF* | 27.4 \| 18.6 | 1930 29 Oct | - | - | 981.9 | 1830 29 Oct |
| | *COAWST* | 26.5 \| 21.5 | 1700 \| 1620 29 Oct | 8.6 | 1750 & 1800 29 Oct | 982.5 | 1810 29 Oct |
| **44009** | *buoy* | 23.7 | 2040 29 Oct | 7.4 | 1050 29 Oct | 956.4 | 2050 29 Oct |
| | *WRF* | 26.8 \| 22.8 | 2320 29 Oct \| 0400 30 Oct | - | - | 963.0 | 0030 30 Oct |
| | *COAWST* | 29.8 \| 25.6 | 0210 30 Oct | 5.7 | 2140 29 Oct | 950.3 | 0010 30 Oct |
| **44013** | *buoy* | 20.4 | 1920 29 Oct | 6.9 | 0150 30 Oct | 988.2 | 0050 30 Oct |
| | *WRF* | 25.0 \| 17.3 | 2030 29 Oct | - | - | 989.2 | 2300 29 Oct |
| | *COAWST* | 24.3 \| 16.3 | 2140 29 Oct | 7.0 | 2210 29 Oct | 989.0 | 2230 29 Oct |
| **44020** | *buoy* | 20.6 | 2000 29 Oct | 3.1 | 1850 29 Oct | 983.3 | 1950 29 Oct |
| | *WRF* | 28.0 \| 19.0 | 1940 29 Oct | - | - | 984.4 | 2040 29 Oct |
| | *COAWST* | 25.9 \| 17.4 | 1930 29 Oct | 2.9 | 1530 29 Oct | 984.7 | 2110 29 Oct |
| **44065** | *buoy* | 24.9 | 0010 30 Oct | 9.9 | 0050 30 Oct | 958.1 | 2150 29 Oct |
| | *WRF* | 30.5 \| 22.6 | 2310 29 Oct \| 0310 30 Oct | - | - | 952.5 | 2310 29 Oct |
| | *COAWST* | 30.9 \| 22.7 | 2200 29 Oct \| 0100 30 Oct | 6.2 | 0100 30 Oct | 960.9 | 2320 29 Oct |

The evaluation of the WRF and COAWST simulations of Hurricane Sandy thus indicates relatively high fidelity.

Nevertheless, the fidelity is lower for simulations of Hurricane Irene and biases relative to observations provide important context for the following analyses which focus on power production and extreme conditions at prospective offshore wind turbine locations. Due to the presence of errors in tropical cyclone tracking in the simulations, in the following discussion of geophysical conditions we consider not only grid cells with wind turbines in the LA, but also ocean-based grid cells nearby.

### 3.2 Wind turbine power production and operating conditions: Hurricane Irene

Mean power production and CF computed for the entire Hurricane Irene simulation period using WRF and COAWST are; $1.51 \times 10^4$ MW (0.38) and $1.56 \times 10^4$ MW (0.39), respectively. When Hurricane Irene is present in d02, equivalent CF are 0.39 and 0.40, respectively. These CF are slightly lower than the climatologically representative estimates of 0.45 presented previously (Pryor and Barthelmie, 2024b, a) due to relatively low wind speeds in the vicinity of the offshore LA early in the simulation and to an extended period of above cut-out wind speeds during the hurricane passage from late on 27 August to

the middle of 29 August (Fig. 5a and Fig. 6a, b). However, the system-wide CF only drops below 0.2 for continuous periods of 5 h 50 min (2000 UTC 28 August through 0150 UTC 29 August) in WRF and 7 h 10 min (1950 UTC 28 August through 0300 UTC 29 August) in COAWST (Fig. 5a). At no point is the projected power production zero.



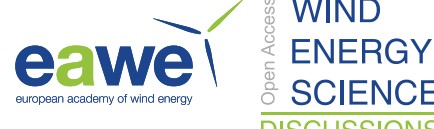

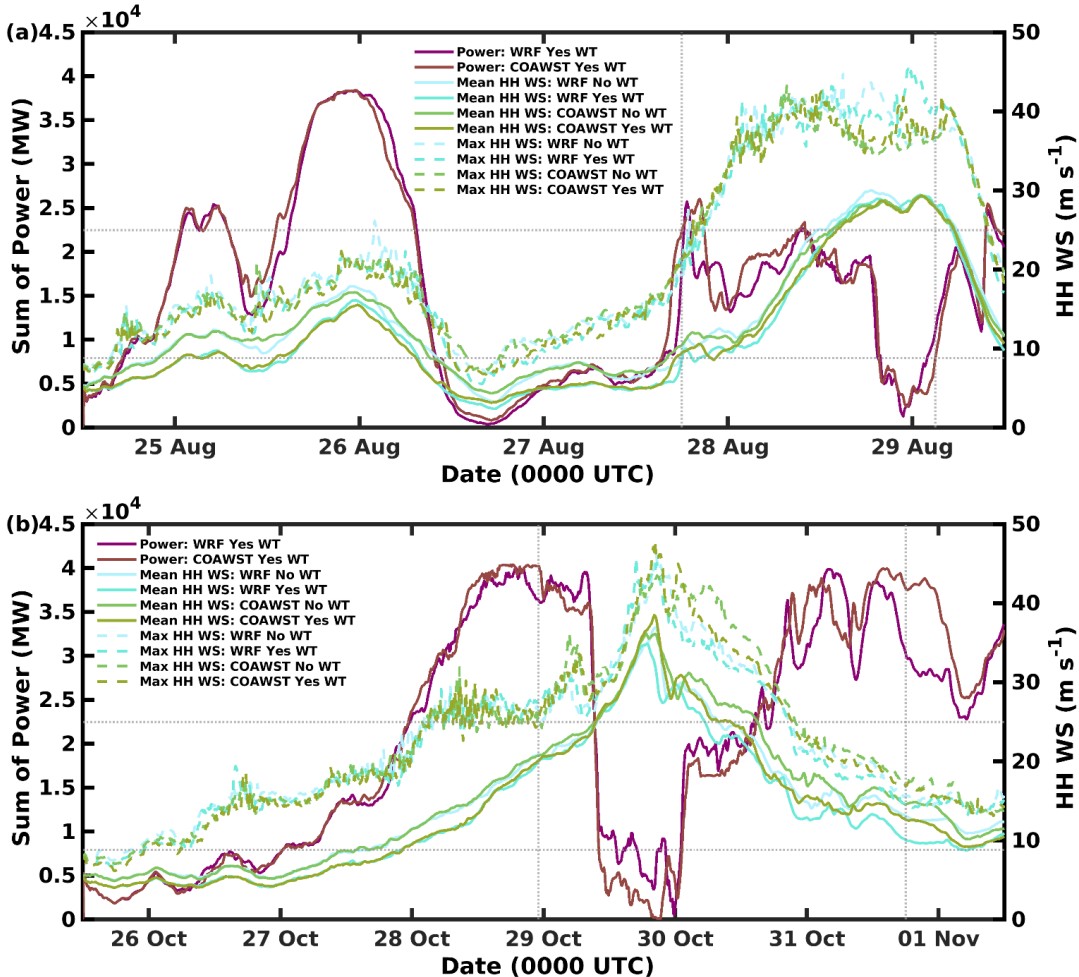

**Figure 5: Time series of total instantaneous power production from all 2642 wind turbines (left axis) and mean and maximum hub-heigh wind speed (HH WS, right axis) in grid cells containing wind turbines for (a) Hurricane Irene and (b) Hurricane Sandy simulations. The dashed gray vertical lines mark the start and end time of storm tracking within d02. The lower dashed gray horizontal line marks power production equivalent to a capacity factor of 0.2, and the upper dashed gray horizontal line marks HH WS = 25 m s⁻¹.**

**Figure 6: Mean wind turbine capacity factor (CF) from (a, c) WRF simulations and (b, d) COAWST simulations for (a, b) Hurricane Irene and (c, d) Hurricane Sandy. Also shown are time series of the CF and mean hub-height wind speed (HH WS) for the four LA clusters. Orange dashed lines indicate the start and end time of storm tracking within d02. The purple dashed line represents the time when the location of the minimum SLP is closest to the cluster center. The red dashed line indicates CF = 0.2, yellow line indicates HH WS of 25 m s⁻¹.**

Time series of power production from WRF and COAWST indicate a high degree of agreement (Fig. 5a) but there are times when the models deviate both in terms of power production and extreme wind speed. Late on 27 August and early on 28 August when Hurricane Irene is south of the LAs, projected power production differs by a maximum of $1.09 \times 10^4$ MW (CF difference of 0.275). With WRF, for the 5 h 50 min when the system-wide CF remains below 0.2, the mean HH WS in WT grid cells for the simulation with the WFP active ranges from 27.6 to 29.3 m s⁻¹. Within the LA the maximum HH WS is 45.4 m s⁻¹, which exceeds the 50-year RP WS at 100 m derived in earlier work using ERA5 output (Barthelmie et al., 2021),



but remains below the 50 m s$^{-1}$ sustained wind speed threshold for class I wind turbines and the 57 m s$^{-1}$ threshold for tropical cyclone hardened wind turbines (class T) (IEC, 2019a). Analyses including all ocean-based grid cells within the four

LA clusters indicate the mean HH WS as simulated by WRF with WFP active ranges from 24.2 to 27.0 m s$^{-1}$ and the maximum HH WS reaches 46.4 m s$^{-1}$. During the period when the system-wide CF from COAWST is < 0.2, the mean HH WS in WT grid cells ranges from 27.2 to 29.3 m s$^{-1}$ and the maximum reaches 41.8 m s$^{-1}$. For all ocean-based grid cells within the four LA clusters, the mean HH WS as simulated by COAWST with the WFP active ranges from 23.1 to 27.0 m s$^{-1}$ and the maximum reaches 42.1 m s$^{-1}$.


Mean HH WS > 25 m s$^{-1}$ in WT grid cells and CF < 0.2 extend for 13.8, 13.3, and 7.2 h in the WRF simulation with the WFP active and 15.2, 12.8, and 7.3 h in the equivalent simulation with COAWST for LA clusters C, B, and A, respectively (Figs. 6 and 7). Due to slight differences in the hurricane tracking (Fig. 1a), the mean CF from COAWST exceeds that with WRF for offshore LA cluster B and in the northern part of LA cluster C, while the mean CF with WRF exceeds that with

COAWST in the southern part of LA cluster C (Fig. 6a, b and Fig. S11). Hurricane Irene tracks very close to LA cluster D, which experiences mean HH WS in WT grid cells > 25 m s$^{-1}$ and CF < 0.2 for 15.5 and 17.3 h in the WRF and COAWST simulations, respectively. According to Mood's test, the median CF for this LA from the WRF and COAWST simulations differ at the 95% confidence level.

Nearly two-thirds of WT-containing grid cells and over three-quarters of ocean grid cells within the LA exhibit a higher frequency of HH WS > 25 m s$^{-1}$ in the COAWST simulations when Hurricane Irene is within d02. A larger $R_{18}$ value is also much more frequent ($\geq$ 96% of time stamps) in each COAWST simulation. Thus, consistent with the analyses of precipitation volume from Hurricane Irene given above, there is evidence that the simulations with COAWST result in a more intense and larger tropical cyclone.


In the COAWST simulation with the WFP active, maximum Hs in WT grid cells within LA clusters A, B, C, and D is; 8.6, 8.5, 7.6, and 7.2 m, respectively (Fig. 8), and thus are all below the 50-year RP Hs of ~ 11 m estimated using the ERA5 dataset (Barthelmie et al., 2021). LA cluster A exhibits the highest frequency (~ 4%) of joint Hs, HH WS, and Tp values (Fig. 8) that fall in the classes centered at Hs $\geq$ 8.4 m, HH WS $\geq$ 35 m s$^{-1}$ (approximately equivalent to 5-m WS of 21.5 m s$^{-1}$

), and Tp $\geq$ 11.2 s that were previously reported to be associated with high mudline bending moments based on simulations with 3D IFORM applied to the 5 MW NREL offshore reference wind turbine (Valamanesh et al., 2015). The COAWST simulation also indicates frequent occurrence of wind-wave misalignment. In the HH WS class 10.6 – 25 m s$^{-1}$, 47, 86, 74, and 32% of the time periods have wind-wave misalignment $\geq$ 30° at the center of LA clusters A, B, C, and D, respectively. For HH WS > 25 m s$^{-1}$, the corresponding values are 22, 41, 44, and 31%, respectively.






**Figure 7: Mean hub height wind speed (HH WS) and time series of the fraction of wind turbine grid cells with HH WS > 25 m s⁻¹**
**(left axis) plus the mean (blue) and maximum (green) HH WS in those grid cells (right axis) in WRF and COAWST simulations with the WFP active. Orange dashed lines indicate the start and end time of storm tracking within d02. The purple dashed line represents the time when the location of the minimum SLP is closest to the cluster center. The yellow line indicates HH WS of 25 m s⁻¹. For simulations without the WFP active, see Fig. S10.**





Figure 8: Extreme conditions based on simulations of Hurricane Irene using COAWST with the WFP active. (a-d) 3-D bubble charts of the joint occurrence of HH WS, Hs, and Tp (5 classes for each variable for a total of 125 possible classes) for all wind turbine grid cells in each LA cluster (A-D). (e-h) Joint probability distributions of HH WS and Hs where the magenta symbols denote 10-min output from all WT grid cells in each LA cluster and the contours denote probability of 0.01 (blue), 0.02 (green), and 0.05 (yellow). (i-l) Histograms of the direction difference (HH WS minus Hs) at the center of each LA cluster for the three HH WS classes: 3 to < 10.6 m s⁻¹, 10.6 to 25 m s⁻¹, and > 25 m s⁻¹.



### 3.3 Wind turbine power production and operating conditions: Hurricane Sandy

Mean instantaneous power production and CF from WRF and COAWST for the entire Hurricane Sandy simulation period are; $2.03 \times 10^4$ MW (0.51) and $2.07 \times 10^4$ MW (0.52), respectively. Considering only the time when Hurricane Sandy is within d02, equivalent CF are 0.62 and 0.61, respectively. The high CF are reflective of high, but below cut-out, HH WS

prior to the passage of the hurricane over the LA and the relatively short duration of HH WS > 25 m s$^{-1}$ within the LA (Fig. 5b and Fig. 6c, d). Simulated system-wide CF drops below 0.2 for 8 h (scattered during 1320 UTC 29 October through 0100 UTC 30 October) in the WRF simulation and for 15 h (1010 UTC 29 October through 0110 UTC 30 October) in the COAWST simulation (Fig. 5b). A single time-step (2100 UTC 29 October) has zero system-wide power production in the COAWST simulation.


During periods when the system-wide CF < 0.2 (including landfall in New Jersey), the mean HH WS in WT grid cells is > 25 m s$^{-1}$ in both the WRF and COAWST simulations with the WFP active (Fig. 5b). In the WRF simulation with the WFP active, during the longest continuous time when the system-wide CF remains below 0.2 (1710 through 2040 UTC 29 October), the mean HH WS is 30.6 to 34.8 m s$^{-1}$ in WT grid cells and 29.7 to 31.9 m s$^{-1}$ in all ocean-based grid cells within

the LA clusters. Equivalent values from COAWST (with WFP active), are 26.0 to 38.5 m s$^{-1}$ and 25.8 to 35.0 m s$^{-1}$.

HH WS > 50 m s$^{-1}$ is simulated in tens of thousands of space-time sample combinations in both the WRF and COAWST simulations with the WFP active. However, none occurred within 170 km of any LA centroid. Maximum HH WS in WT grid cells and the frequency of HH WS > 25 m s$^{-1}$ in WT grid cells is higher (59% and 65% of time stamps when Hurricane

Sandy is within d02) in the COAWST simulation than in the WRF simulation with and without the WFP active (Fig. 7). In all ocean-based grid cells within the LA clusters, 63% and 69% of the time stamps exhibit more grid cells with HH WS > 25 m s$^{-1}$ in the COAWST simulations. Maximum HH WS in ocean-based grid cells within the LA clusters is 45.1 m s$^{-1}$ in WRF and 48.9 m s$^{-1}$ in COAWST (Fig. 7). In the COAWST simulation with the WFP active, the maximum HH WS in WT grid cells in LA clusters A, B, and C are; 44.6, 47.7, and 45.4 m s$^{-1}$, respectively (Fig. 9). They thus exceed the highest 50-year

RP wind speed at 100 m a.s.l. of 39.7 m s$^{-1}$ computed using ERA5 output (Barthelmie et al., 2021), but are below the 50 m s$^{-1}$ and 57 m s$^{-1}$ thresholds for class I and class T wind turbines (IEC, 2019a). Larger $R_{18}$ values prior to landfall are also more frequent in the COAWST simulations (> 70% of time stamps in both the simulations without and with the WFP). Thus, consistent with analyses of the simulations of Hurricane Irene, there is evidence that use of COAWST (for the configuration used herein) results in a larger and more intense hurricane.


Minor differences in the tracking (Fig. 1b) and intensity of the hurricane-induced wind speeds (Fig. 7), causes higher mean CF from LA cluster A and parts of B and C in the simulation with COAWST than the simulation with WRF (Fig. 6 and Fig. S11). The simulations track the centroid of Hurricane Sandy close to LA clusters B and C and accordingly, periods with CF < 0.2 are of greatest duration for these clusters (20.0 and 23.5 h in WRF and 23.3 and 31.0 h in COAWST, respectively)

(Fig. 6). Largest differences in CF are found for LA cluster D. The duration of time with CF < 0.2 is substantially longer in the COAWST simulation due to the prevalence of HH WS > 25 m s$^{-1}$ and the median CF for this cluster between the two simulations differs at the 99% confidence level according to Mood's test.

Maximum Hs of 8.3, 10.4, 7.5, and 6.9 m in LA clusters A, B, C, and D are higher than those for Hurricane Irene but are also

below the 50-year RP values of ~ 11 m derived from ERA5 (Barthelmie et al., 2021) (Fig. 9). LA cluster B exhibits the highest frequency (~ 4%) of joint Hs, HH WS, and Tp values (classes centered at Hs ≥ 8.4 m, HH WS ≥ 35 m s$^{-1}$ [5-m WS ~ 21.5 m s$^{-1}$], and Tp ≥ 11.2 s) close to those associated with a peak mudline moment (of ~ 120 MN-m) (Valamanesh et al.,



2015). Misalignment of wind and waves by ≥ 30° is common for both the HH WS class of 10.6 – 25 m s⁻¹ and > 25 m s⁻¹.

Based on COAWST output from the centroids of LA A, B, C, and D, wind-wave misalignment ≥ 30° is found for; 43 (49),

63 (27), 83 (41) and 34 (49) % of time steps (value in brackets for WS HH > 25 m s⁻¹).

**Figure 9: Extreme conditions based on simulations of Hurricane Sandy using COAWST with the WFP active. (a-d) 3-D bubble charts of the joint occurrence of HH WS, Hs, and Tp (5 classes for each variable for a total of 125 possible classes) for all wind turbine grid cells in each LA cluster (A-D). (e-h) Joint probability distributions of HH WS and Hs where the magenta symbols denote 10-min output from all WT grid cells in each LA cluster and the contours denote probability of 0.01 (blue), 0.02 (green), and 0.05 (yellow). (i-l) Histograms of the direction difference (HH WS minus Hs) at the center of each LA cluster for the three HH WS classes: 3 to < 10.6 m s⁻¹, 10.6 to 25 m s⁻¹, and > 25 m s⁻¹.**





### 3.4 Wind turbine impacts on hurricane properties

Consistent with expectations, removal of kinetic energy by wind turbines means that maximum HH WS in WRF and COAWST simulations with the WFP active tend to be lower than those when the wind turbines are excluded (see Fig. 5). For example, output from the no wind turbine COAWST simulation of Hurricane Irene indicates HH WS in ocean-based grid cells beyond the LA > 50 m s$^{-1}$ 837 times while the corresponding number for the simulation with wind turbines is 333. However, simulations using both WRF and COAWST with full deployment of wind turbines in existing lease areas

(approximately 2600 at an ICD of 4.3 MW km$^{-2}$), indicate that for this scale of offshore wind turbine deployment, the net impact is small expect for hub-height wind speeds near the lease areas.

For all other metrics, the simulations with WRF or COAWST differ more than the simulations with or without the action of wind turbines included (Fig. 10). The 10-min mean precipitation volume within 375 km of the Hurricane Irene centroid

differs (without WT versus with WFP active) by 4.7% and 3.9% in simulations with WRF and COAWST, respectively, but differs (WRF versus COAWST) by 11.8% and 9.2% in the no WT and WT simulations, respectively. The equivalent values for Hurricane Sandy are 8.7% for WRF and 12.9% for COAWST (without WFP active versus with WFP active) and 13.0% for the no WT and 11.6% for the WT simulations (WRF versus COAWST). Similarly, the mean 500 hPa wind speed close to the hurricane centroids differ by < 2 m s$^{-1}$ in simulations of Hurricanes Irene and Sandy with and without WT with no

consistent signal in terms of which simulation is higher (Fig. 10). The mean $R_{18}$ for Hurricane Irene based on WRF simulations with and without the WFP active differ by < 2 km (mean of ~ 280 km). Mean $R_{18}$ from COAWST simulations with and without the WFP active also differ by < 2 km (mean of ~ 300 km). Thus, while median $R_{18}$ from WRF versus COAWST are statistically different (at p < 0.01) for simulations with and without the WFP active, the use of the WFP does not yield significantly different $R_{18}$ values in simulations with a given model. Simulations with WFP active produce equal or

slightly more total precipitation. For Hurricane Irene, the 10-min mean (median) precipitation volume from WRF without and with the WFP active are $1.88 \times 10^8$ m$^3$ ($1.89 \times 10^8$ m$^3$) and $1.88 \times 10^8$ m$^3$ ($1.93 \times 10^8$ m$^3$), while equivalent values from COAWST are $1.93 \times 10^8$ m$^3$ ($1.93 \times 10^8$ m$^3$) and $1.98 \times 10^8$ m$^3$ ($2.02 \times 10^8$ m$^3$), respectively. For Hurricane Sandy, the corresponding values are $8.64 \times 10^7$ m$^3$ ($8.60 \times 10^7$ m$^3$) versus $8.68 \times 10^7$ m$^3$ ($8.10 \times 10^7$ m$^3$) from WRF and $8.69 \times 10^7$ m$^3$ ($9.19 \times 10^7$ m$^3$) versus $8.87 \times 10^7$ m$^3$ ($7.79 \times 10^7$ m$^3$) from COAWST.


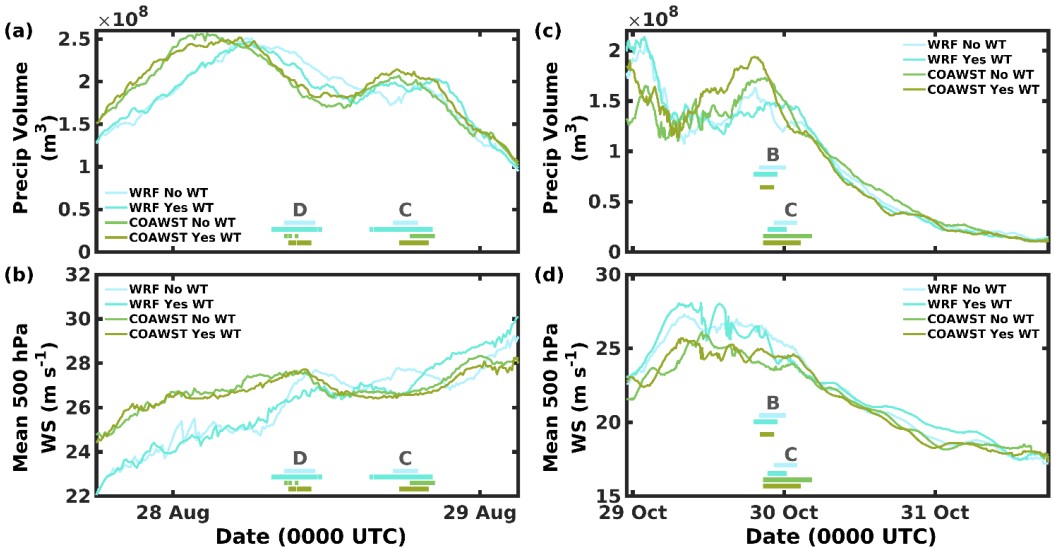

**Figure 10:** Time series of (a, c) 10-min precipitation volume within a 375 km radius from the minimum SLP location and (b, d) mean 500 hPa wind speed of grid cells from 50 to 375 km from the minimum SLP location in d02 for simulations with WRF and COAWST of (a, b) Hurricane Irene and (c, d) Hurricane Sandy. The horizontal lines near "D" and "C" in panels a and b and near "B" and "C" for panels c and d, mark the times when the minimum SLP is within 100 km of the center of the specified offshore wind energy LA cluster. See Fig. S12 for comparison of wind speeds at 10-m a.s.l.

## 4 Concluding remarks

Results of analyses of simulations with WRF and COAWST of two of the most powerful hurricanes that tracked within 100 km of offshore wind energy lease areas along the U.S. East Coast during the last 25 years can be summarized as:

1) Many aspects of Hurricane Sandy are well reproduced in WRF and COAWST control simulations that exclude the action of wind turbines. Consistent with past research, simulations of Hurricane Irene exhibit lower fidelity relative to a range of observations in part due to the negative bias in translational speed. In contrast to similar simulations of Typhoon Muifa (Liu et al., 2015) but consistent with past research on intense cyclones in the North Sea (Larsén et al., 2019), COAWST simulations of both hurricanes indicate a slightly larger area of storm-force wind speeds ($R_{18}$) and hub-height wind speeds > 25 m s$^{-1}$ plus higher precipitation volumes than are indicated by the WRF simulations. This coupled with the ability of COAWST to quantify additional geophysical parameters of importance to offshore structures strongly indicates the need for increasing investment in coupled simulations for the offshore wind energy industry.

2) Despite the intensity and size of these hurricanes and their proximity to the offshore wind energy lease areas, simulations presented herein, that assume a 15 MW reference wind turbine deployed with a spacing of 1.85 km, indicate only fairly brief periods with low power production (system-wide CF < 0.2). System-wide capacity factors below 0.2 due to wide-spread occurrence of hub-height wind speeds above 25 m s$^{-1}$ extend for only 6-7 h in the simulations of Hurricane Irene and 8-15 h for Hurricane Sandy (the longer period is based on the simulation with COAWST). Further, neither hurricane is simulated to produce hub-height wind speeds > 50 m s$^{-1}$ in the current offshore lease areas. Thus, based on these simulations of these intense tropical cyclones there is no evidence of a need for hurricane hardening of wind turbines deployed in the current offshore lease areas. Also, these simulations suggest even such that the projected fleet of offshore wind turbines will continue to supply substantial amounts of



electricity to the grid even during these extreme events. However, simulations of both hurricanes with COAWST result in wave-wind conditions that have previously been identified as being associated with high mudline bending
moments on monopile foundations. The COAWST simulations of both hurricanes also indicate a relative high frequency of HH WS-wave directional misalignment (> 30°) in these lease areas.

    3)  There is no evidence that deployment of 2642 wind turbines at an ICD of 4.3 MW km$^{-2}$ within existing offshore wind energy lease areas along the U.S. East Coast would have substantially weakened either of the hurricanes
600           considered herein. Although much denser and larger deployments might have an influence on hurricanes, even for Hurricane Irene that tracked closest to the offshore wind energy lease areas, simulations with either WRF or COAWST differ more than simulations with either WFP inactive or active with respect to the volume of precipitation near the hurricane center, storm intensity, and/or extent.

Mesoscale simulations performed at convection permitting resolution such as those presented herein allow simulation of the hurricane lifespans and associated power production over large domains and can be used as here to assess whether improved treatment of atmosphere-ocean dynamical coupling alters extreme conditions of relevance to offshore wind turbines. However, it is important to acknowledge that the highest structural loading may occur in the cyclone eye-wall (Han et al., 2014) that is of a scale (Marks et al., 2008) that is not fully represented in the simulations presented here. Nevertheless,
analyses of the simulations suggest the structure of the hurricanes is reasonably represented in our modeling (Fig. 3) and simulations performed at the same grid spacing were shown to represent some aspects of flow in the eye wall (Müller et al., 2024). Future work employing mesoscale-microscale coupling (Wang et al., 2024) could be used to evolve further details of geophysical properties of relevance to structural loading. Further, the hurricanes simulated herein were extremely powerful and both tracked within 100 km of offshore wind energy lease area cluster centers (C and D for Hurricane Irene, B and C for
Hurricane Sandy). However, they do not represent a comprehensive climatology of historical or possible intense tropical/extratropical cyclones (Barthelmie et al., 2021). Undertaking comparable simulations of additional extreme cyclones would also be useful in determining if findings presented herein are generalizable and to quantify the degree to which the meteorological and oceanic extreme conditions vary according to the precise model formulation.

**Code and data availability**

COAWST software can be downloaded from: https://github.com/DOI-USGS/COAWST. NAM data can be downloaded from: https://rda.ucar.edu/datasets/d609000/ and https://www.ncei.noaa.gov/products/weather-climate-models/north-american-mesoscale. OSTIA-UKMO-L4-GLOB-v2.0 SST data can be downloaded from: https://podaac.jpl.nasa.gov/dataset/OSTIA-UKMO-L4-GLOB-v2.0. GHSHHG data can be downloaded from: https://www.ngdc.noaa.gov/mgg/shorelines/. GEBCO data can be downloaded from: https://download.gebco.net/. HYCOM
GLBa0.08 expt 90.9 data can be downloaded from: https://tds.hycom.org/thredds/catalog.html. ADCIRC 2001v2d data can be downloaded from: https://adcirc.org/products/adcirc-tidal-databases/. GFS wind forcing data can be downloaded from: https://www.ncei.noaa.gov/thredds/catalog/model/gfs.html. WW3 data for boundary conditions can be downloaded from: https://www.ncei.noaa.gov/thredds-ocean/catalog/ncep/nww3/catalog.html. NHC "best track" data can be downloaded from: https://www.nhc.noaa.gov/data/tcr/index.php. HURDAT2 data can be downloaded from:
https://www.nhc.noaa.gov/data/#hurdat. IMERG V07 data can be downloaded from: https://disc.gsfc.nasa.gov/datasets/GPM_3IMERGHH_07/summary?keywords="IMERG_final". NDBC buoy data can be downloaded from: https://www.ndbc.noaa.gov/. Scientific color maps can be downloaded from https://www.fabiocrameri.ch/colourmaps/ (Crameri et al., 2020).



**Supplemental materials**

See the attached document.

**Author contributions**

SCP and RJB obtained funding and computing support for this work. KBT performed the simulations, SCP and KBT processed data used for model evaluation, conducted the analyses with assistance from RJB, produced the figures, and drafted the initial manuscript. All authors contributed to the design of the analyses, interpretation of results, and manuscript
revisions.

**Competing interests**

The authors declare that they have no conflict of interest.

**Acknowledgements**

This research was supported by the Cornell Atkinson Center for Sustainability (AVF-AOWE) and the U.S. Department of
Energy Office of Science (DE-SC0016605). Computing resources were provided by NSF Extreme Science and Engineering Discovery Environment (XSEDE) (award TG-ATM170024) and the National Energy Research Scientific Computing Center, a DoE Office of Science User Facility supported by the Office of Science of the U.S. Department of Energy under Contract No. DE-AC02-05CH11231. Any opinions, findings, and conclusions or recommendations expressed in this material are those of the authors and do not necessarily reflect the views of these agencies. We thank Dr. John Warner for his help in
answering COAWST-related questions.

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
