# Peer review of "Hurricane impacts in the United States East Coast offshore wind energy lease areas"

_Wind Energy Science, 2025_

## Referee Comment (RC1)

**Reviewer comments on "Hurricane impacts in the United States East Coast offshore wind energy lease areas"**

April 4, 2025

**1 General comments**

The manuscript "Hurricane impacts in the United States East Coast offshore wind energy lease areas" by Thompson et al. presents simulations of two hurricane cases along the U.S. East Coast. Four model configurations are analyzed for each hurricane case, including atmosphere-only WRF simulations and atmosphere-wave-ocean coupled COAWST simulations. In addition, the effects of wind turbines on the hurricanes are analyzed using simulations with and without wind farm parameterizations.

The manuscript contains novel aspects such as 1.) the use of buoys and IMERG data to validate tropical cyclone simulations, 2.) the presented analysis of the atmosphere-ocean-wave-coupled modeling framework applied to tropical cyclone cases including wind turbine effects. The manuscript addresses three clearly stated research questions that are relevant to the wind energy sector and of broad international interest. The methods are well described and the analysis is valid. The title and abstract give a good summary of the manuscript, the manuscript is well written and overall well structured.

The manuscript uses eleven rather long tables, and not all of them may be needed in the main article. Many figures show panels for both WRF and COAWST simulations, although in some cases the difference between the two is not easy to see (see specific suggestions in the Specific Comments section). At the same time, some figures contain a lot of information while being rather small.

**2 Specific comments**

1. Introduction: Lines 34-40: You might consider extending this argument for tropical cyclone events, and include that tropical cyclones may not be adequately covered by available offshore measurements.

2. Line 184: Note that the effective model resolution using WRF is about 7 times the grid spacing Skamarock (2004).

3. Figure 1a,b): I think it would be helpful to the reader to show these two panels larger in the main article. This would help to better see the agreement between the observed and simulated track, and you could also consider not showing Table 1. The precipitation time series is also shown in Fig. 10, I think one of the two figures should be sufficient.

4. Fig. 2b and Table 2), the scheme of COAWST is not further explained in the text. So I would suggest not to show Fig. 1 b and Table 2b in the main article, but to give them in the supplement.

5. line 255: Could you explain what $3 \times 3$ smoothing means?

6. Figures 1, 3, 6, 7: The difference between the WRF and COAWST simulations is not easy to see in these figures. While the difference between the two is discussed to some extent in the text, one could consider not showing both panels in all cases, or showing the difference between the two. For Figures 6 and 7, you might consider showing the map and wind field for either the COAWST or WRF simulation only, while showing the time series for both in the same plot.

7. Figures 8, 9: It is difficult to read the Tp, HH WS, and Hs from the 3-D bubble plots. Can you consider a different visualization; this could be adding the Tp via contours or colors in the plots (e-h), or showing a separate joint probability of HH WS and Tp.

**3   Technical corrections**

1. Line 21: The abbreviation HH WS is used only once in the abstract. Therefore, I would recommend not introducing it.

2. Line 28: The abbreviation IC is used only once in the abstract. I would recommend introducing abbreviations only if they are used more than once.

3. Line 135: Personally, I have never come across the term "storyline simulations" and would prefer to use "case study" instead.

4. Section 2.1: You could refer to Fig. 1a and b when describing the hurricanes.

5. For the date format, e.g. in Table 1 and others, following the mathematical notation and terminology guidelines of wind energy science, I would recommend using "27 Aug 18:00" instead of "1800 27 Aug".

6. Line 179: I would suggest referring to Fig. 1c in a separate sentence, explicitly stating that the figure shows the power and thrust curves used for the Fitch parameterization.

7. Table 3: The table shows not only the sources of the initial and boundary conditions but also the model resolution; could you update the title?

8. Line 150: The abbreviation WS has already been introduced in line 58.

9. Line 278: I think you meant to write "of 3 - 10.6 $\mathrm{m\,s}^{-1}$,..."?

10. Line 275: I suggest introducing the peak period with a few more words, e.g. "period of the peak energy in the wave spectrum".

11. Table 5, 6: To help the reader, you might consider removing the horizontal lines between the buoy, WRF, and COAWST data of the same buoy location.

12. Fig. 5 and others: While the idea of grouping the colors is good, the colors between "COAWST No WT" and "COAWST Yes WT" and between "WRF No WT" and "WRF Yes WT" are too close together to be clearly distinguished.

**References**

Skamarock, W. C.: Evaluating mesoscale NWP models using kinetic energy spectra, Mon. Weather Rev., 132, 3019–3032, https://doi.org/10.1175/MWR2830.1, 2004.

---

## Author Response (AR1)

Comments from the reviewer are shown in black and our modifications are given in blue.

We express thanks to the reviewer for their thoughtful comments as we note in the acknowledgements.

Reviewer comments on "Hurricane impacts in the United States East Coast offshore wind energy lease areas"

April 4, 2025

**1 General comments**

The manuscript "Hurricane impacts in the United States East Coast offshore wind energy lease areas" by Thompson et al. presents simulations of two hurricane cases along the U.S. East Coast. Four model configurations are analyzed for each hurricane case, including atmosphere-only WRF simulations and atmosphere-wave-ocean coupled COAWST simulations. In addition, the effects of wind turbines on the hurricanes are analyzed using simulations with and without wind farm parameterizations.

The manuscript contains novel aspects such as 1.) the use of buoys and IMERG data to validate tropical cyclone simulations, 2.) the presented analysis of the atmosphere-ocean-wave-coupled modeling framework applied to tropical cyclone cases including wind turbine effects. The manuscript addresses three clearly stated research questions that are relevant to the wind energy sector and of broad international interest. The methods are well described and the analysis is valid. The title and abstract give a good summary of the manuscript, the manuscript is well written and overall well structured.

Many thanks for your positive comments and assessment.

The manuscript uses eleven rather long tables, and not all of them may be needed in the main article. Many figures show panels for both WRF and COAWST simulations, although in some cases the difference between the two is not easy to see (see specific suggestions in the Specific Comments section). At the same time, some figures contain a lot of information while being rather small.

We fully accept that the manuscript as submitted contained an atypically large number of tables (6) in addition to the 10 figures.

We provide details of our changes to Figures below. With respect to Tables:

- 1) Moved Table 1 to Supplemental Materials (Table S1)
- 2) Moved Table 2 to Supplemental Materials (Table S2)
- 3) Integrated Table 4 into the text. Thus, the text at the start of Section 2.2 changed from: "In this research, both WRF (v4.2.2) and COAWST (v3.7 and MCT v2.6.0) simulations use two domains (Fig. 2a) and the coupling interval in COAWST is 10 min (Fig. 2b). The source of boundary and initial conditions and key physics options (Tables 3 and 4) are informed by previous simulations of Hurricanes Sandy (Zambon et al., 2014b) and Irene (Mooney et al., 2016). The MYNN2 planetary boundary layer scheme is used due to the compatibility with the Fitch windfarm parameterization (WFP) (Fitch et al., 2012) that is

used here to compute power production, momentum extraction, and turbulent kinetic energy (TKE) induced by the action of wind turbines."

**To:**

"In this research, both WRF (v4.2.2) and COAWST (v3.7 and MCT v2.6.0) simulations use two domains (Fig. 2a) and the coupling interval in COAWST is 10 min. At this coupling interval, a number of variables that are critical to air-sea coupling and lower atmosphere structure and/or WT design standards are exchanged between the model components (Fig. 2b, Fig. S3, and Table S2). The selection of these variables is based on previous research (Warner et al., 2010; Zambon et al., 2014b) and include sea surface temperature (SST) that is passed from ROMS to WRF, 10 m u- and v-wind components which are passed from WRF to SWAN, plus Hs and Tp (period of peak energy in the wave spectrum) that are passed from SWAN to WRF and ROMS. The source of boundary and initial conditions (Table 1) and key physics options are informed by previous simulations of Hurricanes Sandy (Zambon et al., 2014a) and Irene (Mooney et al., 2016). Physics settings include the WRF single-moment 6-class (WSM6; (Hong and Lim, 2006)) microphysics scheme, the Rapid Radiative Transfer Model (RRTM; (Mlawer et al., 1997)) for longwave radiation, the Dudhia scheme (MM5; (Dudhia, 1989)) for shortwave radiation, and the Unified Noah land surface model (Chen and Dudhia, 2001b, a; Ek et al., 2003; Tewari et al., 2004). The Kain-Fritsch (Kain, 2004) cumulus parameterization is used in the outer domain and no cumulus parameterization is used in the inner domain. The Mellor-Yamada Nakanishi and Niino Level 2.5 (MYNN2; (Nakanishi and Niino, 2006)) planetary boundary layer scheme is used due to the compatibility with the Fitch windfarm parameterization (WFP; (Fitch et al., 2012)) that is used here in both domains to compute power production, momentum extraction, and turbulent kinetic energy (TKE) induced by the action of WTs."

**2 Specific comments**

1. Introduction: Lines 34-40: You might consider extending this argument for tropical cyclone events, and include that tropical cyclones may not be adequately covered by available offshore measurements.

Yes, quite. We have modified the text to read:

"The offshore environment presents significant challenges for making long-term, climatologically representative robust measurements of properties such as wind speed at WT hub-height (HH WS) (Foody et al., 2024) that are critical for determining the wind resource and key aspects of operating conditions (IEC, 2019b, a; Mudd and Vickery, 2024). The relative paucity of measurements leads to financial uncertainty and thus potentially jeopardizes realizing national goals for achieving the energy transition (Hansen et al., 2024). It also means that numerical modeling is playing a critical role in projecting wind resource and operating conditions in offshore wind energy development areas (Kresning et al., 2020; Pryor and Barthelmie, 2021; Bodini et al., 2024; Pryor and Barthelmie, 2024a; Wang et al., 2024). Limited over-ocean observations also limit our ability to characterize the characteristics of high intensity hurricanes, including those of

relevance to the wind energy industry, particularly in environments such as the U.S. East Coast which has the potential to be impacted by tropical cyclones and/or transitioning tropical-extratropical cyclones (Xie et al., 2005; Baldini et al., 2016; Barthelmie et al., 2021; Wang et al., 2024) but experiences only relatively few such storms each century (Schreck III et al., 2021)."

2. Line 184: Note that the effective model resolution using WRF is about 7 times the grid spacing Skamarock (2004).

Agreed. We have changed this sentence:

"Note the wind speeds output from d02 are for a nominal model time step of 2 s but are representative of a spatial average of 1.33 km by 1.33 km, while the design standards are for a sustained wind speed at a point (Larsén and Ott, 2022).

**To read:**

"Note it is not an expectation that spatially averaged model output will perfectly match time-averaged point observations and further, the design standards are articulated for a sustained WS at a point (Larsén and Ott, 2022). The WSs presented here are output from d02, represent a nominal model time step of 2 s, and are from a grid cell with an area of 1.33 km by 1.33 km, but the effective model resolution is ~ 7 times the grid spacing (Skamarock, 2004) thus any spatial gradients will be under-estimated."

3. Figure 1a,b): I think it would be helpful to the reader to show these two panels larger in the main article. This would help to better see the agreement between the observed and simulated track, and you could also consider not showing Table 1. The precipitation time series is also shown in Fig. 10, I think one of the two figures should be sufficient.

The hurricane tracks in panels (a) and (b) from Figure 1 have become a separate figure (new Figure 1) and the other figure panels are now their own figure (new Figure 3, see below). Table 1 is now located in the Supplemental Materials (Table S1) and key information from that table is now included in the Figure 1 caption (see below).

Figure 10 and Figure 1 show slightly different time series. Figure 10 shows the 10-min precipitation volume within a 375 km radius of the minimum SLP and Figure 1 (now Figure 3) shows the 1-h precipitation volume within a 375 km radius of the minimum SLP.

Modified Figure 1

Modified Figure 3

4. Fig. 2b and Table 2), the scheme of COAWST is not further explained in the text. So I would suggest not to show Fig. 1 b and Table 2b in the main article, but to give them in the supplement.

We apologize – the text we provided was too brief. The first paragraph of Section 2.2 that began:

"The source of initial and lateral boundary conditions (Khaira and Astitha, 2023) and specific model configurations employed within WRF and COAWST (including the coupling system) have a critical impact on simulated flow conditions (Mooney et al., 2019). In this research, both WRF (v4.2.2) and COAWST (v3.7 and MCT v2.6.0) simulations use two domains (Fig. 2a) and the coupling interval in COAWST is 10 min (Fig. 2b)."

is now expanded to include this further sentence:

"At this coupling interval, a number of variables that are critical to air-sea coupling and lower atmosphere structure and/or WT design standards are exchanged between the model components (Fig. 2b, Fig. S3, and Table S2). The selection of these variables is based on previous research (Warner et al., 2010; Zambon et al., 2014b) and include sea surface temperature (SST) that is passed from ROMS to WRF, 10 m u- and v-wind components which are passed from WRF to SWAN, plus Hs and Tp (period or peak energy in the wave spectrum) that are passed from SWAN to WRF and ROMS."

Panel (b) from what was Figure 2 has become Figure S3 and Table 2 has become Table S2. A new panel (b) has been included with only variables discussed in the main text and is shown below.

Modified Figure 2

5. line 255: Could you explain what  $3 \times 3$  smoothing means?

**This sentence:**

"Hurricane centroid locations are computed every 10 minutes as the minimum SLP after 3x3 smoothing is applied to the model output and are used for comparison with the NHC best track information."

**has been modified to read:**

"Hurricane centroid locations are computed every 10 min as the minimum SLP after 3×3 smoothing is applied to the model output (a mean value of SLP is computed for each grid cell based on output for that grid cell and the eight adjacent grid cells) and compared with the NHC best track information."

6. Figures 1, 3, 6, 7: The difference between the WRF and COAWST simulations is not easy to see in these figures. While the difference between the two is discussed to some extent in the text, one could consider not showing both panels in all cases, or showing the difference between the two. For Figures 6 and 7, you might consider showing the map and wind field for either the COAWST or WRF simulation only, while showing the time series for both in the same plot.

Please see (3) above for changes to Figure 1. With Figures 3, 6, and 7 (now Figures 4, 7, and 8), the COAWST panels remain in the main text; all four (WRF and COAWST) panels are included in the Supplemental Materials (Figures S4, S12, and S13). With Figure 7, the time series plots now mark when the WRF CF exceeds the corresponding COAWST CF by > 0.05 in brown and when the COAWST CF exceeds the corresponding WRF CF by > 0.05 in green. With Figure 8, the time series plots now mark when the WRF HH WS exceeds the corresponding COAWST HH WS by > 0.5 m s-1 in gray and when the COAWST HH WS exceeds the corresponding WRF HH WS by > 0.5 m s-1 in magenta.

Modified Figure 7

Modified Figure 8

7. Figures 8, 9: It is difficult to read the Tp, HH WS, and Hs from the 3-D bubble plots. Can you consider a different visualization; this could be adding the Tp via contours or colors in the plots (e-h), or showing a separate joint probability of HH WS and Tp.

3D plots are indeed inherently tricky. We did evaluate a range of different options before selecting bubble plots and frankly found the bubble plots to be the best option given; Tp is categorial (integer seconds) and contouring in 3D space is hard particularly when the data are highly "concentrated" – e.g., panel b and noting we wanted to preserve information regarding the co-occurrence of the three variables.

- 3 Technical corrections
- 1. Line 21: The abbreviation HH WS is used only once in the abstract. Therefore, I would recommend not introducing it.

With apologies for our error, the abbreviation HH WS has been removed from the abstract.

2. Line 28: The abbreviation IC is used only once in the abstract. I would recommend introducing abbreviations only if they are used more than once.

Thank you for pointing this out, we have corrected this.

3. Line 135: Personally, I have never come across the term "storyline simulations" and would prefer to use "case study" instead.

"Storyline simulations" as a concept has evolved in the climate science community (see discussion in Doblas-Reyes, F. J., and Coauthors, 2021: Linking global to regional climate change. Climate Change 2021: The Physical Science Basis, V. Masson-Delmotte et al., Eds., Cambridge University Press, 1363–1512.). In brief, the difference between a case study and a storyline is that a case study is purely a geophysical event while a storyline inherently is a geophysical event that is "impact" or decision maker relevant. For this reason, we prefer the term storyline since we are explicitly considering these hurricanes in the context of risk to the renewable energy sector.

4. Section 2.1: You could refer to Fig. 1a and b when describing the hurricanes.

Section 2.1, "Characteristics of the hurricanes considered herein", is a good location to refer to panels (a) and (b) of Figure 1 (which are now a separate figure – as noted in the third comment in the previous section) – this has been added.

**Section 2.1 now reads:**

"Research presented herein focuses on two recent hurricanes:

- 1) Hurricane Irene became a category 3 hurricane, with 54 m s-1 WSs at 10 m height in the Bahamas on 24 August 2011 12:00 UTC (Avila and Cangialosi, 2011). It made landfall at Cape Lookout, North Carolina on 27 August 12:00 UTC with 39 m s-1 10 m WSs. After moving out over the water, it again made landfall, this time as a tropical storm, with 31 m s-1 WSs reported at Brigantine, New Jersey on 28 August 2011 09:35 UTC (Fig. 1a). The cyclone then moved over Coney Island, New York with 28 m s-1 WSs reported at 13:00 UTC. Simulations presented herein are initialized on 24 August 2011 12:00 UTC and run through 29 August 2011 12:00 UTC.
- 2) Hurricane Sandy became a category 3 hurricane, with 51 m s-1 WSs at 10 m height in eastern Cuba on 25 October 2012 05:25 UTC (Blake et al., 2013; Lackmann, 2015). It grew to have a roughly 1611 km diameter of tropical-storm-force WSs, before making landfall near Brigantine, New Jersey as a post-tropical cyclone with 36 m s-1 10 m WSs and a minimum pressure of 945 hPa on 29 October 2012 23:30 UTC (Fig. 1b). Simulations presented herein run from 25 October 2012 12:00 UTC through 1 November 2012 12:00 UTC."
- 5. For the date format, e.g. in Table 1 and others, following the mathematical notation and terminology guidelines of wind energy science, I would recommend using "27 Aug 18:00" instead of "1800 27 Aug".
  - Thank you for highlighting the guidelines for date and time. Throughout the manuscript, "dd month yyyy, hh:mm UTC" formatting is now used.
- 6. Line 179: I would suggest referring to Fig. 1c in a separate sentence, explicitly stating that the figure shows the power and thrust curves used for the Fitch parameterization.
  - Reworded to read: "Following previous research (Pryor and Barthelmie, 2024a, b), we assume that all auctioned offshore LAs along the U.S. East Coast (Fig. 2a) are populated with 2642 IEA reference 15 MW WTs, each of which has a hub height of 150 m, and a rotor diameter of 240 m (see power and thrust curves in Fig. 2c), at a spacing of 1.85 km for an average ICD of 4.3 MW km-2."
- 7. Table 3: The table shows not only the sources of the initial and boundary conditions but also the model resolution; could you update the title?
  - The title for Table 3 (now Table 1) has been changed from "Sources of initial and boundary conditions for WRF and COAWST" to "Model configuration for WRF and COAWST simulations".

8. Line 150: The abbreviation WS has already been introduced in line 58.

The duplicate explanation of wind speeds (WS) has been removed with our apologies.

9. Line 278: I think you meant to write "of 3 - 10.6 ms-1,..."?

Sorry for any confusion. To make the three class ranges more clear, "to" has replaced "–": The sentence now reads "... in HH WS classes of 3 to  $< 10.6 \text{ m s}^{-1}$ ,  $10.6 \text{ to } 25 \text{ m s}^{-1}$ , and  $> 25 \text{ m s}^{-1}$ , to represent ..." and follows the format in the caption and legend of Fig. 8 (now Fig. 9) and Fig. 9 (now Fig. 10). The sentences on lines 452 and 523 have been modified to also use "to" instead of "–" (10.6 to 25 m s-1).

10. Line 275: I suggest introducing the peak period with a few more words, e.g. "period of the peak energy in the wave spectrum".

Done

11. Table 5, 6: To help the reader, you might consider removing the horizontal lines between the buoy, WRF, and COAWST data of the same buoy location.

Table 5 and Table 6 (now Table 2 and Table 3) have been modified to remove the horizontal lines between the buoy, WRF, and COAWST data of the same buoy location.

12. Fig. 5 and others: While the idea of grouping the colors is good, the colors between "COAWST No WT" and "COAWST Yes WT" and between "WRF No WT" and "WRF Yes WT" are too close together to be clearly distinguished.

We want to make sure that everyone can clearly distinguish between the different colors. We have chosen a new color scheme (see below) that is also suggested in the Crameri et al. (2020) reference and is now used with the updated figures. Additional line styles are now also used in Figure 6 (previously Figure 5).

Modified Figure 6

**References**

Skamarock, W. C.: Evaluating mesoscale NWP models using kinetic energy spectra, Mon. Weather Rev., 132, 3019–3032, https://doi.org/10.1175/MWR2830.1, 2004.

Comments from the reviewer are shown in black and our modifications are given in blue.

We express thanks to the reviewer for their thoughtful comments as we note in the acknowledgements.

Reviewer comments on "Hurricane impacts in the United States East Coast offshore wind energy lease areas"

This study is one of the first to use a coupled atmosphere-ocean-wave model to study the interactions between hurricanes and offshore wind turbines. I think this is very relevant as the deployment region is often subject to hurricanes. The research questions are very clear and are addressed by the results. However, I think the structure of the results section could be improved. In addition, the inclusion of a more recent roughness length parameterization of the wind wave alignment could strengthen the study's alignment with the stated research objectives.

**Major comments:**

One of the main objectives of this paper is to identify high wind wave misalgnments, which is essential for understanding structural loading. Therefore, I propose to use a wind-wave aware roughness length parameterization for a more accurate representation of this process. For example, Fu et al. (2023) show that including such a parameterization improves wind estimates, which I assume is important for this study. An alternative could be the parameterization presented by Porchetta et al. (2019), which has also shown improved hub height wind speeds compared to older schemes. Integrating either of these would likely increase the relevance and impact of the current work.

Fu, S., Huang, W., Luo, J., Yang, Z., Fu, H., Luo, Y., and Wang, B. (2023) Deep leaning-based sea surface roughness parameterization scheme improves sea surface wind forecast. Geophysical Research Letters, 50(24), e2023GL106580. <a href="https://doi.org/10.1029/2023GL106580">https://doi.org/10.1029/2023GL106580</a>

Porchetta, S., Temel, O., Munoz-Esparza, D., Reuder, J., Monbaliu, J., van Beeck, J. and van Lipzig, N. (2019) A new roughness length parameterization accounting for wind-wave (mis)alignment. Atmospheric Chemistry and Physics, 19(10), 6681–6700. https://doi.org/10.5194/acp-19-6681-2019

Naturally there is a need for further exploration of alternative model configurations, including alternative roughness length parameterizations. Such a comprehensive analysis is beyond the scope of this work but would be extremely valuable. We explicitly note this in the conclusions sections where we write: "Undertaking comparable simulations of additional extreme cyclones and simulations with different configurations including alternative  $z_0$  parameterizations (Porchetta et al., 2019; Fu et al., 2023) and a wave boundary layer model within SWAN (Du et al., 2017) would also be useful in determining if findings presented herein are generalizable and to quantify the degree to which the meteorological and oceanic extreme conditions vary according to the precise model formulation."

It may be helpful to separate the results and discussion sections, as the current layout makes it difficult to follow. I also recommend improving the structure within the results section. It currently includes comparisons between two hurricanes, multiple models (WRF (WFP), COAWST (WFP)), and different parameters, which makes it dense and sometimes inconsistent. Consider focusing on the main results and moving supporting but non-essential material to the appendix. I also suggest rethinking the figures and their layout - while the content is valuable, the presentation makes it hard to digest. Emphasizing the differences between model results or including bias/RMSE metrics may improve clarity.

We regret that the results section is difficult to follow. Section 3 "Results" has been renamed "Results and discussion". The section is structured to address the three numbered objectives from section 1.2 in order, and chronologically (i.e., Irene then Sandy) within each of the objectives. Subsection names and numbers have been modified to make the order clearer: 3.1 Evaluation of simulations without the action of wind turbines, 3.1.1 Hurricane Irene, 3.1.2 Hurricane Sandy, 3.1.3 Synthesis, 3.2 Wind turbine power production and operating conditions, 3.2.1 Hurricane Irene, 3.2.2 Hurricane Sandy, and 3.3 Wind turbine impacts on hurricane properties. By including discussion points within the results, we hope to address comparisons to previous studies, highlight important features, etc., as soon as they relate to the results as opposed to including duplicate text from the results in a separate section prior to the inclusion of discussion points.

Figure 1 has been separated into two separate figures. See "Figure 1:" below for additional details. Figures 3, 6, and 7 (now 4, 7, and 8) now only show two of the four panels – those for COAWST. For CF (Fig. 7), the time series plots show the respective COAWST values, and now also highlight when the WRF CF exceeds the corresponding COAWST CF by > 0.05 in brown and when the COAWST CF exceeds the corresponding WRF CF by > 0.05 in green. For HH WS (Fig. 8), the time series plots show the respective COAWST values, and now also highlight when the WRF HH WS exceeds the corresponding COAWST HH WS by > 0.5 m s-1 in gray and when the COAWST HH WS exceeds the corresponding WRF HH WS by > 0.5 m s-1 in magenta. Four panel plots with both WRF and COAWST are now located in Supplemental Materials. Please see Figs. 7 and 8 below.

Modified Figure 7

Modified Figure 8

**Minor comments:**

Line 10: Please specify what is meant by "high resolution" in this context.

We have reworded this paragraph to read: "Four sets of high-resolution simulations are performed for two category 3 tropical cyclones that tracked close to current offshore wind energy lease areas to assess the possible impacts on, and from, wind turbines. Simulations of Hurricanes Irene and Sandy are performed at convective permitting resolution (grid spacing in inner domain of 1.33 km) with both the Weather Research and Forecasting model (WRF, v4.2.2) and the Coupled Ocean-Atmosphere-Wave-Sediment Transport (COAWST, v3.7) model to characterize geophysical conditions of relevance to offshore wind turbines."

Line 13: Could you add the version numbers of the models used?

The sentence has changed as noted above

Figure 1: The figure is hard to interpret. The plots do not speak for themselves - please clarify what is being shown (e.g. sum or difference of precipitation) and make it more readable without relying solely on the caption.

Figure 1 has been separated into two figures. The new Figure 1 includes the hurricane tracks and the new Figure 3 includes panels containing precipitation. With Figure 3, the colorbar descriptions are now "Accumulated Precip (mm): IMERG", "Precip (mm): WRF No WT Minus IMERG", and "Precip (mm): COAWST No WT Minus IMERG".

Line 110: Could you explain why the wave boundary layer model was not used in your setup?

There are indeed many options and we stuck with formulations close to those that had been previously used but have noted in the conclusions; "Undertaking comparable simulations of additional extreme cyclones and simulations with different configurations including alternative  $z_0$  parameterizations (Porchetta et al., 2019; Fu et al., 2023) and a wave boundary layer model within SWAN (Du et al., 2017) would also be useful in determining if findings presented herein are generalizable and to quantify the degree to which the meteorological and oceanic extreme conditions vary according to the precise model formulation."

Table 2: Consider removing this table if it is not essential to the main results.

Table 2 has been moved to Supplemental Materials (Table S2).

The following sentences have been modified and added to the first paragraph of Section 2.2: "At this coupling interval, a number of variables that are critical to air-sea coupling and lower atmosphere structure and/or WT design standards are exchanged between the model components

(Fig. 2b, Fig. S3, and Table S2). The selection of these variables is based on previous research (Warner et al., 2010; Zambon et al., 2014b) and include sea surface temperature (SST) that is passed from ROMS to WRF, 10 m u- and v-wind components which are passed from WRF to SWAN, plus Hs and Tp (period or peak energy in the wave spectrum) that are passed from SWAN to WRF and ROMS."

Line 240: It may be worthwhile to briefly mention the limitations or uncertainties of the evaluation data sets used.

The following has been added to the end of Section 2.3 Evaluation data sets:

"These data sets do have some inherent constraints, which include use of; subjective smoothing to produce representative 6 h best track data which does not necessarily equate to a precise storm history (Landsea and Franklin, 2013), spatial averaging on the gridded IMERG data which can underestimate high precipitation rates compared to point measurements (Hu and Franzke, 2020; Nie and Sun, 2020; Huffman et al., 2024), and the limited number and spatial coverage of buoys (NDBC, 2009)."

Line 255: What is meant by "3x3 smoothing"? Please clarify.

**This sentence:**

"Hurricane centroid locations are computed every 10 minutes as the minimum SLP after 3x3 smoothing is applied to the model output and are used for comparison with the NHC best track information."

**has been modified to read:**

"Hurricane centroid locations are computed every 10 min as the minimum SLP after 3×3 smoothing is applied to the model output (a mean value of SLP is computed for each grid cell based on output for that grid cell and the eight adjacent grid cells) and compared with the NHC best track information."

Line 298: This section seems to mix results and discussion - consider separating them for better flow.

Please see the above reply in the "Major Comments" section.

Line 300: The evaluation here is quite dense, with several variables and metrics presented at once. A clearer structure for comparisons would help.

With Section 3, subsections have been added and renamed to provide more clarity.

- 3.1 Evaluation of simulations without the action of wind turbines
- 3.1.1 Hurricane Irene
- 3.1.2 Hurricane Sandy

- 3.1.3 Synthesis
- 3.2 Wind turbine power production and operating conditions
- 3.2.1 Hurricane Irene
- 3.2.2 Hurricane Sandy
- 3.3 Wind turbine impacts on hurricane properties.

Line 300: It's hard to see this clearly in Figure 1a. How is "fidelity" defined in this context?

We have modified this sentence to read: "As shown in detail below, simulations of Hurricane Irene exhibit lower fidelity than those of Hurricane Sandy."

Line 304: Can bias be quantified and presented in a table?

We have sought to clarify this in the text rather than adding another table.

Line 313: What does "R18" refer to? Please define.

We do define it where we write: "The mean outermost radius of tropical storm force WSs at 10 m ( $R_{18}$ ,  $18 \text{ m s}^{-1}$ , Fig. 4) is computed using azimuth sectors of  $10^{\circ}$  (Powell and Reinhold, 2007) for all sectors where the distance from the cyclone centroid to the d02 boundary is  $\geq 200 \text{ km}$  and used as a measure of cyclone size."

Lines 496-510: Could the observed changes in wind speed be related to differences in roughness length? This may be worth investigating as it may help explain some of the results.

Possibly, though we did not find a clear/definitive signal in local z0.

Line 541: Should this be "except"?

Thank you for pointing this out. It has been corrected.

Line 574: It is unclear which model setup provides better hurricane estimates - please clarify.

We have added this information in section 3.1.3:

"Evaluation of the WRF and COAWST simulations of Hurricane Sandy thus indicates relatively high fidelity. Nevertheless, the fidelity is lower for simulations of Hurricane Irene and biases relative to observations provide important context for the following analyses which focus on power production and extreme conditions at prospective offshore WT locations. Due to the presence of errors in tropical cyclone tracking in the simulations, in the following discussion of geophysical conditions we consider not only grid cells with WTs in the LAs, but also oceanbased grid cells nearby. In terms of agreement with; observed precipitation, cyclone size (R18), near-surface WS and cyclone tracking, COAWST simulations exhibit higher skill than those with WRF."

We base this assessment on the following summary of information presented in the manuscript: Irene:

- WRF precipitation within 300 km range of centroid better agrees with IMERG.
- COAWST R18 better agrees with HURDAT2
- COAWST exhibits better agreement with buoy observations of near-surface wind speeds
   Sandy
  - WRF and COAWST comparable agreement in terms of precipitation within 300 km range of centroid relative to IMERG
  - COAWST better agreement in terms of centroid location v HURDAT2
  - COAWST better agreement with buoy observations of near-surface wind speeds

Line 590: Be consistent in terminology when referring to hurricanes versus cyclones.

The sentence has been changed from: "Thus, based on these simulations of these intense tropical cyclones there is no evidence of a need for hurricane hardening of wind turbines deployed in the current offshore lease areas."

to: "Thus, based on these simulations of these intense hurricanes there is no evidence of a need for hurricane hardening of WTs deployed in these LAs."

Could input files be provided in order for others so that they can repeat the work if necessary.

This information was given in Supplemental Materials but we have now also added the link to a persistent repository that provides input files to the "Code and data availability" Section.

---

## Author Response (AR2)

**Report #1**

| Submitted on 03 Jun 2025 |
|--------------------------|
| Anonymous referee #1     |

**Checklist for reviewers**

ATTENTION: before filling this section, please check if are you reviewing a normal submission or a data description article. For a normal submission, please fill the top part of the form entitled "Non-data description articles". For data description articles, fill out the relevant part of the form entitled "Data description articles". PLEASE DO NOT FILL BOTH PARTS.

**Non-data description articles**

All types of manuscripts (except for data description articles) need to be evaluated by you according to the following three criteria.

**1) Scientific significance:**

Does the manuscript represent a substantial contribution to scientific progress within the scope of Wind Energy Science (substantial new concepts, ideas, methods, analyses, or data)?

**2) Scientific quality:**

Are the scientific approach and applied methods valid? Is sufficient information given so other researchers (in principle) can repeat the work? Are the results discussed in an appropriate and balanced way (consideration of related work, including appropriate references)?

**3) Presentation quality:**

Are the scientific results and conclusions presented in a clear, concise, and well-structured way (abstract conveys efficiently the essence of the paper; number and quality of figures/tables; appropriate, fluent and precise use of English language)?

**Data description articles**

For **data description articles**, you are asked not just to assess the manuscript but, more importantly, the data set itself. If you are reviewing another manuscript type, please select "n/a" for the questions below.

**Excellent Good Fair Poor n/a**

Excellent Good Fair Poor n/a

Excellent Good Fair Poor n/a

**1) Scientific significance: Is there any potential of the data being useful? This is clearly the most important decision. There are at least three subcriteria to evaluate: **Uniqueness**: it should not be possible to replicate the Excellent Good Fair Poor n/a experiment, observation or data generation on a routine basis. This is also the case for cost-intensive data sets that might not be replicated due to economic reasons. Usefulness: it should be plausible that the data, alone or in Excellent Good Fair Poor n/a combination with other data sets, can be used in future investigations, for the comparison to model outputs or to verify other experiments or observations. Completeness: a data set must not be intentionally split, for Excellent Good Fair Poor n/a example, to increase the possible number of publications. It should contain all data that can be reviewed without unnecessary increase of workload and that can be reused in another context by a reader. 2) Data quality: Excellent Good Fair Poor n/a The data must be ready and accessible for inspection and analysis to make the reviewer's task possible. Even if a submitted data set is the first ever published, its claimed accuracy, the instrumentation employed, and methods of processing should reflect the "state of the art" or the current "best practices". Reviewers will then apply their expert knowledge and experience to perform tests (e.g. statistical tests) and judge whether the data and any possible claimed findings are plausible and do not contain detectable faults. 3) Presentation quality: Excellent Good Fair Poor n/a The article should describe in a clear, concise and wellstructured way the data set and how it was obtained, using an appropriate, fluent, and precise use of the English language. The article text and references should contain all information necessary to evaluate all claims about the data set, whether the claims are explicitly written down in the article, or implicit, through the data being published or their metadata. The authors should point to suitable software or services for simple and free visualization and analysis. For final publication, the manuscript should be accepted as is. accepted subject to technical corrections. accepted subject to minor revisions.**

reconsidered after major revisions:

rejected.

**Were a revised manuscript to be sent for another round of reviews: I would be willing to review the revised manuscript.**

I would not be willing to review the revised manuscript.

**Suggestions for revision or reasons for rejection**

(visible to the public if the article is accepted and published)
* * *
Response: We thank the reviewer for their original comments and this positive assessment.

**Report #2**

Submitted on 04 Jul 2025 Anonymous referee #2 **Checklist for reviewers** ATTENTION: before filling this section, please check if are you reviewing a normal submission or a data description article. For a normal submission, please fill the top part of the form entitled "Non-data description articles". For data description articles, fill out the relevant part of the form entitled "Data description articles". PLEASE DO NOT FILL BOTH PARTS. Non-data description articles All types of manuscripts (except for data description articles) need to be evaluated by you according to the following three criteria. 1) Scientific significance: Excellent Good Fair Poor n/a Does the manuscript represent a substantial contribution to scientific progress within the scope of Wind Energy Science (substantial new concepts, ideas, methods, analyses, or data)? 2) Scientific quality: Excellent Good Fair Poor n/a Are the scientific approach and applied methods valid? Is sufficient information given so other researchers (in principle) can repeat the work? Are the results discussed in an appropriate and balanced way (consideration of related work, including appropriate references)? 3) Presentation quality: Excellent Good Fair Poor n/a Are the scientific results and conclusions presented in a clear, concise, and well-structured way (abstract conveys efficiently the essence of the paper; number and quality of figures/tables; appropriate, fluent and precise use of English language)? **Data description articles** For data description articles, you are asked not just to assess the manuscript but, more importantly, the data set itself. If you are reviewing another manuscript type, please select "n/a" for the questions below. 1) Scientific significance: Is there any potential of the data being useful? This is clearly

| the most important decision. There are at least three sub-
criteria to evaluate:                                                                                                                                                                                                                                                                                                                                                                                                                                                                                         |                              |  |
|-----------------------------------------------------------------------------------------------------------------------------------------------------------------------------------------------------------------------------------------------------------------------------------------------------------------------------------------------------------------------------------------------------------------------------------------------------------------------------------------------------------------------------------------------------------------------------|------------------------------|--|
| Uniqueness : it should not be possible to replicate the experiment, observation or data generation on a routine basis. This is also the case for cost-intensive data sets that might not be replicated due to economic reasons.                                                                                                                                                                                                                                                                                                                                      | Excellent Good Fair Poor n/a |  |
| Usefulness : it should be plausible that the data, alone or in combination with other data sets, can be used in future investigations, for the comparison to model outputs or to verify other experiments or observations.                                                                                                                                                                                                                                                                                                                                           | Excellent Good Fair Poor n/a |  |
| Completeness : a data set must not be intentionally split, for example, to increase the possible number of publications. It should contain all data that can be reviewed without unnecessary increase of workload and that can be reused in another context by a reader.                                                                                                                                                                                                                                                                                             | Excellent Good Fair Poor n/a |  |
| 2) Data quality: The data must be ready and accessible for inspection and analysis to make the reviewer's task possible. Even if a submitted data set is the first ever published, its claimed accuracy, the instrumentation employed, and methods of processing should reflect the "state of the art" or the current "best practices". Reviewers will then apply their expert knowledge and experience to perform tests (e.g. statistical tests) and judge whether the data and any possible claimed findings are plausible and do not contain detectable faults.          | Excellent Good Fair Poor n/a |  |
| 3) Presentation quality: The article should describe in a clear, concise and well-structured way the data set and how it was obtained, using an appropriate, fluent, and precise use of the English language. The article text and references should contain all information necessary to evaluate all claims about the data set, whether the claims are explicitly written down in the article, or implicit, through the data being published or their metadata. The authors should point to suitable software or services for simple and free visualization and analysis. | Excellent Good Fair Poor n/a |  |
| For final publication, the manuscript should be                                                                                                                                                                                                                                                                                                                                                                                                                                                                                                                             |                              |  |
| accepted as is.                                                                                                                                                                                                                                                                                                                                                                                                                                                                                                                                                             |                              |  |
| accepted subject to technical corrections.                                                                                                                                                                                                                                                                                                                                                                                                                                                                                                                                  |                              |  |
| accepted subject to minor revisions.                                                                                                                                                                                                                                                                                                                                                                                                                                                                                                                                        |                              |  |
| reconsidered after major revisions: rejected.                                                                                                                                                                                                                                                                                                                                                                                                                                                                                                                               |                              |  |
| rejected.                                                                                                                                                                                                                                                                                                                                                                                                                                                                                                                                                                   |                              |  |

**Were a revised manuscript to be sent for another round of reviews:**

I would be willing to review the revised manuscript.

I would not be willing to review the revised manuscript.

**Suggestions for revision or reasons for rejection**

(visible to the public if the article is accepted and published)

The reviewers' efforts to enhance the readability of the text and figures are appreciated. I would like to offer a few concluding observations.

It is unfortunate that the authors are unwilling to explore an alternative roughness parameterization scheme, especially since such a scheme has been shown to better predict wind fields under wind—wave misalignment, which is directly related to the research question of this manuscript. Considering that the simulations are relatively short, incorporating an additional scheme would likely have been feasible within the time frame.

If this is truly outside the scope of the current project, it is imperative to acknowledge the limitations of the present model setup in addressing wind—wave misalignment at an earlier stage in the manuscript. This acknowledgement should be made prior to any mention of future work. Furthermore, the discussion section would benefit from a more extensive examination of the limitations of the model. This should encompass not only the roughness parameterization but also the uncertainty associated with wind farm parameterization.

**Specific remarks:**

Line 71: Please clarify that this refers to wave height.

Line 95: Consider adding "during hurricane events" to clarify the context, similar to the phrasing in line 108 referring to extreme events.

Line 118: Rephrase the storyline for clarity.

Could the uncertainty of the observations be addressed or quantified, at least qualitatively?

**Figure suggestions:**

To improve readability, consider adding titles such as "Irene" and "Sandy" above the respective columns.

**Response:**

It is not really our "unwillingness" to make new simulations with other roughness length schemes. It is an unfortunate reality of finite and fully exhausted resources (both personnel and computational) that has become more acute in recent months.

We note, in case the reviewer missed it, that we did discuss this matter in section 2.2 (much before the discussion of the simulations):

"Variation of wave state and  $z_0$  with WS is an important determinant of extreme, near surface WSs and turbulence intensity (Zambon et al., 2014b; Porchetta et al., 2019; Porchetta et al., 2020; Porchetta et al., 2021; Wang et al., 2024). The COAWST simulations are configured using the Taylor Yelland formulation (Taylor and Yelland, 2001) to calculate  $z_0$  following past research (Zambon et al., 2014a) that found use of this parameterization resulted in better fidelity for Hurricane Sandy track, intensity, SST, and Hs than alternatives (Oost et al., 2002; Drennan et al., 2005). Use of the MYNN surface layer with WRF and the DRAGLIM\_DAVIS drag limiter option with COAWST, means all simulations implement a maximum ocean roughness drag coefficient of  $2.85 \times 10^{-3}$ , consistent with research that has shown asymptotic behavior of drag at high WSs (Davis et al., 2008)."

**Naturally we fully acknowledge:**

- Fu et al. were able to achieve greater agreement with observations of z0 with an ANN model trained using; surface wind speed (WSPD), the angle between the peak wave direction and the wind direction (DIR\_wav\_wind), wave speed (Cp), friction velocity (u\*), significant wave height (Hs), wind direction (DIR\_wind), and peak wave direction (DIR\_wav) than with any physics-based scheme, which is interesting but it is not a formulation that has yet been adopted within the WRF community and could not readily be adopted for our research. Though we note the great potential for machine-learning emulators.
- That the work of Porchetta et al yielded very interesting results in terms of the impact of wind-wave misalignment on surface roughness length. The version of COAWST we employed does not (yet) have this roughness length scheme available as an option.

Accordingly, we have added additional text to the paragraph from 2.2 so it now reads: "Variation of wave state and  $z_0$  with WS is an important determinant of extreme, near surface WSs and turbulence intensity (Zambon et al., 2014a; Porchetta et al., 2019; Porchetta et al., 2020; Porchetta et al., 2021; Wang et al., 2024). Further, wind-wave misalignment also plays a key role in near-surface WSs and  $z_0$  (Porchetta et al., 2019) and machine learning tools have suggested  $z_0$  prediction accuracy can be improved by inclusion of wind-wave directional misalignment as a predictor (Fu et al., 2023). The COAWST simulations are configured using the Taylor Yelland formulation (Taylor and Yelland, 2001) to calculate  $z_0$  following past research (Zambon et al., 2014b) that found use of this parameterization resulted in better fidelity for Hurricane Sandy track, intensity, SST, and Hs than alternatives (Oost et al., 2002; Drennan et al., 2005). Use of the MYNN surface layer with WRF and the DRAGLIM\_DAVIS drag limiter option with COAWST, means all simulations implement a maximum ocean roughness drag coefficient of 2.85 ×  $10^{-3}$ , consistent with research that has shown asymptotic behavior of drag at high WSs (Davis et al., 2008)."

We have also added this comment (last sentence below) at line 260:

"Three-dimensional and joint occurrences of HH WS, Hs, and Tp, in WT-containing grid cells from the COAWST simulations are presented along with histograms of estimated wind-wave misalignment at the LA cluster centers in HH WS classes of 3 to  $< 10.6 \text{ m s}^{-1}$ ,  $10.6 \text{ to } 25 \text{ m s}^{-1}$ , and  $> 25 \text{ m s}^{-1}$ , to represent high thrust, moderate thrust, and above rated

WS (Fig. 2c). We caution that the specific model set-up may play a critical role in dictating wind-wave misalignment and feedback via the  $z_0$  (Porchetta et al., 2019)."

We have also expanded the discussion regarding uncertainty, so the final paragraph of the manuscript now reads:

"Mesoscale simulations performed at convection permitting resolution such as those presented herein allow simulation of the hurricane lifespans and associated power production over large domains and can be used as here to assess whether improved treatment of atmosphere-ocean dynamical coupling alters extreme conditions of relevance to offshore WTs. However, it is important to acknowledge that results from any numerical simulations are subject to uncertainty. For example, the highest structural loading may occur in the cyclone eye-wall (Han et al., 2014) which is of a scale (Marks et al., 2008) that is not fully represented in the simulations presented here. Nevertheless, analyses of the simulations suggest the structure of the hurricanes is reasonably represented in our modeling (Fig. 4 and Fig. S4) and simulations performed at the same grid spacing were shown to represent some aspects of flow in the eye wall (Müller et al., 2024). Further, while the Fitch WFP used herein is the most widely adopted within the wind energy community, such schemes do not comprehensively treat WT wake generation (Fischereit et al., 2022). Past research has suggested that use of alternative schemes such as the Explicit Wake Parameterization tends to lead to substantially lower wake loses in large offshore wind farms and higher power production than are simulated using the Fitch scheme (Pryor and Barthelmie, 2024a). Future work employing mesoscale-microscale coupling (Wang et al., 2024) could be used to evolve further details of geophysical properties of relevance to structural loading. Further, the hurricanes simulated herein were extremely powerful and both tracked within 100 km of offshore wind energy LA cluster centers (C and D for Hurricane Irene, B and C for Hurricane Sandy). However, they do not represent a comprehensive climatology of historical or possible intense tropical/extratropical cyclones (Barthelmie et al., 2021). Undertaking comparable simulations of additional extreme cyclones and simulations with different configurations including alternative  $z_0$  parameterizations (Porchetta et al., 2019; Fu et al., 2023) and a wave boundary layer model within SWAN (Du et al., 2017) would also be useful in determining if findings presented herein are generalizable and to quantify the degree to which the meteorological and oceanic extreme conditions vary according to the precise model formulation."

**Line 71: Please clarify that this refers to wave height.**

Done – The manuscript now reads "Equivalent estimates of extreme WSs and Hs from buoy measurements are; 32.6 m s-1 and 9.5 m, respectively (Kresning et al., 2024)."

Line 95: Consider adding "during hurricane events" to clarify the context, similar to the phrasing in line 108 referring to extreme events.

Done – The manuscript now reads "Only limited previous research has sought to quantify the degree to which wind-wave coupling improves simulation fidelity and/or intensity for WSs at heights of relevance to offshore WTs particularly during extreme events."

**Line 118: Rephrase the storyline for clarity.**

We gather that the reviewer does not like the term storylines. We have rewritten to avoid it. The manuscript now reads "Research presented herein focuses on simulations of two of the most powerful hurricanes that have occurred within the U.S. eastern coastal waters in which offshore wind energy LAs have been auctioned (Fig. 1, see further details in Table S1 and Figs. S1-S2)."

Could the uncertainty of the observations be addressed or quantified, at least qualitatively? This matter was not raised in the prior review, but we are happy to address it. We have added text to the paragraph that begins at line 225 as shown below:

"These data sets have some inherent constraints. For the HURDAT2 data set this includes use of subjective smoothing to produce representative 6 h track data which does not necessarily equate to a precise storm history (Landsea and Franklin, 2013). HURDAT2 accuracy for landfalling hurricanes has been estimated as  $\sim 2.5 \text{ m s}^{-1}$  for maximum WS, 1 hPa for SLP, and the location is correct within approximately  $0.1^{\circ}$  of latitude and longitude (Landsea and Franklin, 2013). Spatial averaging naturally impacts the spatial distribution of precipitation within the gridded IMERG data and can lead to underestimation of high precipitation rates compared to point measurements (Hu and Franzke, 2020; Nie and Sun, 2020; Huffman et al., 2024). With respect to the NDBC data, there is a limited number and thus spatial coverage of buoys (NDBC, 2009). Efforts to optimize buoy design to enhance NDBC measurement accuracy have been previously documented (Taft et al., 2009) as have data quality control procedures (NDBC, 2023). Best available information suggests the total sensor accuracy for WS is  $\pm 1 \text{ m s}^{-1}$ , although lower accuracy may arise during high wave states (NDBC, 2023). For Hs the stated accuracy is  $\pm 0.2 \text{ m}$  and for SLP it is  $\pm 1 \text{ hPa}$ ."

**Figure suggestions:**

To improve readability, consider adding titles such as "Irene" and "Sandy" above the respective columns.

The name of each hurricane is now included in all figures.